# Verbal manipulations of learning expectancy do not enhance reconsolidation

**Julia Marinos, Olivia Simioni, Andrea R. Ashbaugh** *

School of Psychology, University of Ottawa, Ottawa, Ontario, Canada

* andrea.ashbaugh@uottawa.ca

## Abstract

### Objectives

Pharmacological studies using propranolol suggest that if reactivation signals that new information will be learned (i.e., there is an expectation for learning) reconsolidation can be enhanced. We examined if the verbal instructions to expect new learning will enhance reconsolidation of fear memories using the post-retrieval extinction paradigm.

### Methods

On day one, participants (n = 48) underwent differential fear conditioning to two images (CS+ and CS-). On day two, participants were randomly assigned to one of three groups; groups one and two had their memory for the CS+ reactivated (i.e., a single presentation of the CS+) 10 minutes prior to extinction, whereas group three did not have their memory reactivated but went right to extinction (no reactivation group). One reactivation group was told that they would learn something new about the images (expectation for learning group), and the other group was told that they would not learn anything new (no expectation for learning group). On day three, return of fear was measured following reinstatement (i.e., four shocks). Fear potentiated startle (FPS) and skin conductance response (SCR) were measured throughout.

### Results

There was evidence of fear acquisition for participants for SCR but not FPS. With regards to reconsolidation, SCR increased for the CS+ and CS-in all groups from the end of extinction to the beginning of re-extinction (i.e., return of fear). For FPS, post-hoc tests conducted on the sub-group of participants showing fear learning showed that FPS remained stable in the two reactivation groups, but increased to the CS+, but not the CS- in the no reactivation group.

### Implications

These findings suggest that a verbal manipulation of the expectation for learning may not be salient enough to enhance reconsolidation. Results are discussed in relation to theories on

---

**Data Availability Statement:** Although the authors cannot make their study's data publicly available at the time of publication, all authors commit to make the data underlying the findings described in this study fully available without restriction to those

who request the data, in compliance with the PLOS Data Availability policy. For data sets involving personally identifiable information or other sensitive data, data sharing is contingent on the data being handled appropriately by the data requester and in accordance with all applicable local requirements.

**Funding:** This research was supported by an NSERC Discovery Grant (RGPIN-2015-05379) awarded Andrea R Ashbaugh.

**Competing interests:** The authors have declared that no competing interests exist.

differences in between SCR, as a measure of cognitive awareness, and FPS, as a measure of fear.

## Introduction

Reconsolidation is the process where a long-term memory, once reactivated, returns to a malleable state and can be updated [1], strengthened [2], or blocked [3, 4]. This process of memory reconsolidation has been observed in a number of species [5, 6], including humans [4]. Researchers have used various methods to study reconsolidation, including pharmacological techniques, transcranial magnetic stimulation, and behavioural interventions.

Although much of the initial research on reconsolidation was conducted in animals using pharmacological agents, more recent pharmacological studies have been performed on humans using the beta-adrenergic receptor blocker propranolol [7]. Such studies in humans have demonstrated that a conditioned fear response, as measured by fear potentiated startle (FPS), can be eliminated if reactivation is paired with oral administration of propranolol [4, 8–10], though some studies have failed to replicate this effect [11–13]. Of clinical relevance, Brunet et al. [14] demonstrated that physiological responding (i.e., heart rate and skin conductance responses (SCR)) to personalized script driven trauma imagery was reduced one week after the administration of propranolol post recall of the traumatic event compared to a placebo group. These findings were replicated in open-label studies [15, 16]. Reconsolidation has also been incorporated into the treatment of the specific phobia of spiders using propranolol. Soeter and Kindt [10] examined if blocking reconsolidation in individuals with high levels of spider fear would inhibit the return of fear following reinstatement using a behavioural approach test (BAT) as a measure of fear. Participants who had their memory reactivated (i.e., by focusing on their fears) followed by an oral dose of propranolol displayed greater approach behaviour during a BAT than participants who received a placebo or did not have their memory reactivated. Differences between these groups were exhibited at all follow-up periods (i.e., 16 days, three months, and 1 year later) demonstrating that the reduction of fear following reconsolidation persisted long-term. The above studies [4, 8–10, 14–16] illustrate that reconsolidation of memories can be blocked by the administration of propranolol followed by reactivation of the memory and this process might be beneficial in the treatment of anxiety based psychological disorders.

However, there are a number of limitations to the use of propranolol to study memory reconsolidation. First, though animal studies have used protein synthesis inhibitor drugs, such as anisomycin, which directly target reconsolidation mechanisms, anisomycin is toxic to humans [17]. As such, beta-adrenergic receptor blockers, such as propranolol, which modulate, but do not fully block, protein synthesis and therefore do not directly target reconsolidation mechanisms [17] have been used to study reconsolidation in humans. Second, research has found that individuals have a preference for behavioural interventions over drug therapy for the treatment of anxiety disorders [18]. This is important when considering the translational impact of reconsolidation research for the treatment of anxiety disorders and posttraumatic stress disorder. Therefore, there is a need to examine memory reconsolidation using nonpharmacological methods.

Researchers have used a variety of different methods to enhance memory reconsolidation, including behavaioural paradigms and TMS [19–21]. We focus here behavioural paradigms. Schiller et al. [1] examined memory reconsolidation in a human sample using a behavioural

design. They theorized that when extinction follows reactivation, reconsolidaiton rather than new learning would be activated. In their study, a conditioned fear memory was reactivated by presenting the conditioned stimuli without the unconditioned stimuli, followed 10 minutes later by extinction. Consistent with this theory, Schiller et al. [1] found that participants who had their memory reactivated and 10 minutes later underwent extinction, did not display a conditioned fear response as measured by skin conductance following reinstatement. Conversely, participants who only underwent extinction or who had their memory reactivated and then underwent extinction outside the reconsolidation window (i.e., more 6 hours later) exhibited a return of conditioned fear as measured by increased skin conductance following reinstatement 24 hours later. These results have been replicated [22–24] however; several other studies have failed to reproduce these results [4, 8, 9, 25]. Furthermore, a recent registered replication study based upon Schiller's work failed to replicate their results [26]. Additionally, researchers have also highlighted that Schiller et al. excluded a large number of participants and that using different exclusion criteria affected the outcome of their findings [26]. The inconsistencies in replicating these findings suggest that recalling a memory is not always sufficient to reactivate a memory when behavioural methods are used and therefore it is important to develop a better understanding of the conditions under which reconsolidation occurs.

The exploration of the boundary conditions of reconsolidation, or the physiological, environmental, or psychological variables that allow for successful memory reconsolidation [27], is crucial to our understanding of the mechanisms of reconsolidation and its potential translation into clinical interventions. Some research suggests that simply recalling a memory may not enough for the memory to return to a labile state and undergo reconsolidation [28, 29]. For example, Asthana et al. [24] found that individual genetic variations of brain derived neurotrophic factor (BDNF), specifically BDNF *val66met*, may impact the degree to which reconsolidaiton can be enhaced using behavioural methods.

Another boundary condition that has been examined using pharamacological studies is the expectation for learning. The expectation for learning refers to the anticipation that new learning about a contingency is going to occur. Enhancing the expectation for learning by asking participants to explicitly learn about the new contingency has been shown to enhance extinction of a conditioned fear better than extinction without such instructions [30].

Learning expectancy also appears to be relevant, not just for new learning, but also the reconsolidation of previously learned material. When there is a violation between expectation and actual events, referred to as a prediction error, both initial learning and extinction learning is strengthened [31]. As such, researchers have examined whether the expectation for new learning enhances reconsolidation by creating a prediction error [31–35]. Animal research supports the notion that prediction error enhances reconsolidation using pharmacological blockade methods to study reconsolidation [34]. Sevenster et al. [32] examined if a prediction error was needed in order for a memory to return to a labile state and undergo reconsolidation in humans. When there was a prediction error (i.e., reactivation signaled new information) the conditioned fear response did not return following reinstatement, as demonstrated by an elimination of the startle response. In contrast, when there was no prediction error (e.g., reactivation presented the exact same information as the previous day) fear did return following reinstatement as demonstrated by an elevated fear potentiated startle response following reinstatement. These findings suggest that when reactivation signals that something new can be learned (i.e., prediction error) the conditioned fear memory is more likely to be rendered labile and undergo reconsolidation.

Researchers have also examined whether simply inducing the expectancy for learning can trigger reconsolidation by manipulating participants' expectation to receive a shock prior to reactivation using propranolol in humans [35]. In this study, the shock expectancy group,

participants were connected to the shock equipment during reactivation but did not receive a shock as expected. Therefore, a mismatch occurred when the CS+ was presented in the absence of a shock. In the no shock expectancy group, there was no expectation to receive a shock when the CS+ was presented because participants were not connected to the shock equipment during reactivation. As predicted, the group that expected to receive a shock demonstrated reconsolidation, whereas the group that had no expectation for receiving the shock did not demonstrate reconsolidation. These studies illustrate that the expectation for learning during reactivation appears to be critical to the reconsolidation of conditioned fear memories using pharmacological methods.

Overall, research has demonstrated that recalling a memory is not sufficient for a memory to undergo reconsolidation [4, 8, 9, 25]. Pharmacological blockade studies have found that reactivation should indicate that something new can be learned in order for a memory to return to a malleable state and undergo reconsolidation [32, 35]. However, few studies utilizing behavioural experiments (i.e., post-retrieval extinction paradigm) have investigated how the expectation for learning impacts reconsolidation. Huang et al. [36] exposed mice to a novel environment (i.e., creating a prediction error) prior to a retrieval-extinction session of a remote fear memory and found that novelty facilitated fear memory reconsolidation in a behavioural paradigm. Junjiao et al. [37] examined the role of prediction error using similar methods to Sevenster et al. [32] and also found that prediction error was essential for memory destabilization. Furthermore, stronger prediction errors may result in more strongly destabilize memories, as demonstrated by the reduced spontaneous recovery and reinstatement following post-retrieval extinction in participants who received a strong prediction error compared to participants who received a weaker prediction error, who only demonstrated reduced reinstatement only [38]. However, to the best of our knowledge research have not examined if other methods to induce expectancy violation, such as verbal instructions, can facilitate reconsolidation using behavioural methods.

The purpose of the current study was to examine if the expectation for learning–that is, participants' expectation that they were going to be provided with new information about the previously learned association—prior to memory reactivation would impact the reconsolidation process using the post-retrieval extinction paradigm. Participants were randomly assigned to a no-reactivation, a reactivation with expectation for learning, or a reactivation with no expectation for learning condition. Similar to Sevenster et al. [35], the level of expectancy for learning was manipulated by providing each group with different instructions about the relationship between the CS+ and US prior to reactivation. We predicted that participants that had their memory reactivated and expected to learn something new would not display a return of fear following reinstatement on day three (i.e., there would be no change in participants' SCR or FPS response from the end of extinction on day two to the beginning of re-extinction following reinstatement on day three). Conversely, we predicted that the participants who did not expect to learn something new and participants that did not receive reactivation would demonstrate a return of fear on day 3 following reinstatement as demonstrated by increased SCR and FPS.

## Methods

All procedures were approved by the University of Ottawa's Research Ethics Board and all participants provided informed consent.

### Participants

Exclusion criteria included: a self-reported heart condition (e.g., heart transplant, artificial cardiac pacemakers; cardiac arrhythmias, uncontrolled hypo- or hypertension, myocardial

infarction); or reported current use of a beta-blocker. We collected data until we had enough participants with usable physiological data. Ninety-nine undergraduate students from the University of Ottawa were recruited through an online participant pool run by the School of Psychology. Of the 99 participants, the following were excluded: 33 participants dropped out; six individuals were not invited back after day one because they could not identify which image was paired with the shock; and 12 were excluded because their physiological data were excessively noisy overall based on visual inspection (e.g., due to poor electrode contact) or evidence of clipping (e.g., data that were outside of the recording range). The final sample consisted of 48 participants. Participants were compensated with partial one course credit at the end of day one and day two, for a total of two credits, and received $5 for their participation on day three.

## Materials

**Conditioned stimuli.** The conditioned stimuli (CS) consisted of two different images of spiders selected from the International Affective Picture System [IAPS images 1200 and 1201; 39]. The selected IAPS images have been demonstrated to be emotionally arousing in a student population [39]. During fear conditioning on day one, one of the images was paired with the shock 75% of the time (CS+) and the other image was never paired with the shock (CS-). The image associated with CS+ and CS- was counterbalanced across participants.

**Unconditioned stimulus (US).** The US consisted of an electric shock. The shock was delivered by a Grass SD9 Square Pulse Stimulator via two disposable (3.81 x 2.54cm) Ag/AgCl sensors (pre-applied with 0% chloride wet gel) attached to the wrist of the dominant hand. Two 2-meter touchproof snap leads were attached to the sensors and the leads were plugged into the Grass SD9 Square Pulse Stimulator. Before testing, participants determined the level of shock used throughout the three days. The shock was administered starting at 10 volts and increased by 2.5 volts until the participant determined the shock was uncomfortable but not painful up to 60 volts. The same voltage was used on all three days of testing.

## Measures

**The Spider Phobia Questionnaire [SPQ; 40].** The SPQ is a 31-item self-report questionnaire measuring fear of spiders. Participants endorsed either true or false to indicate if each item applies to them. The SPQ has demonstrated acceptable test-retest reliability and discriminant validity in a student sample and can differentiate between participants with and without a specific phobia [41]. This measure was used to ensure level of spider fear was similar across all three groups. Cronbach's alpha was α = .91, for the current sample.

**Spielberger State Trait Anxiety Inventory [42].** The STAI consists of two 20-item self-report questionnaires that assess trait (i.e., STAI-T) and state (i.e., STAI-S) anxiety. The STAI-T asks participants to rate how anxious they generally feel on a 4-point scale ranging from *almost never* (1) to *almost always* (4). The STAI-S asks participants to rate how anxious they feel right now on a 4-point scale ranging from not at all (1) to very much so (4). Both scales have demonstrated adequate convergent validity and excellent test-retest reliability [43]. This measure was used to assess if there were differences in trait and state anxiety levels between groups because high levels of anxiety have been demonstrated to impair reconsolidation [44]. Cronbach's alpha was .92 for the STAI-T and .92 for the STAI-S in the current sample.

**Manipulation check.** Participants were asked at the end of the study before debriefing to rate how much they were expecting to receive a shock at the beginning of day 2 on a scale of 0 (not at all) to 5 (very much) to determine if we were successful in manipulating the expectation for learning among the different conditions. The manipulation check was administered

retrospectively on day three of testing so as not interfere with the reactivation of the memory, as previous research suggests that self-reported assessments of distress (e.g., SUDS ratings) may interfere with reconsolidation [45].

## Physiological measure

**Skin conductance response.** SCR was recorded with BioLab Acquisition Software 3.1.13 from MindWare Technologies Ltd throughout the study. SCR was measured in micro-siemens and sampled constantly at 1000 Hz. The leads were connected to a 16-Channel Electrode Box and the signal was amplified with a Galvanic Skin Conductance Amplifier from Mindware Technologies Ltd. Participants had two disposable (3.81 x 2.54cm) Ag/AgCl sensors (pre-applied with 0% chloride wet gel) placed on the palm of their non-dominant hand on the thenar eminence and hypothenar eminence and two 2-meter touchproof snap leads were connected to the sensors. Leads were taped on the skin with hypoallergenic surgical tape to reduce movement which can interfere with the recording of the data. All physiological measures were recorded on a separate computer from the visual stimuli to ensure the data is recorded with minimal impediment as the computer can overload when both applications are run and can create additional noise in the physiological data.

Data were preprocessed in the EDA analysis software (version 3.1.5) from Mindware Technologies Ltd. We applied a rolling filter with a block size of 100 samples to smooth the data and reduce the influence of noise. All data were visually inspected and rejected when excessive noise distorted the signal (e.g., movement, poor electrode contact). A SCR was defined as a peak minimum of 0.02 $\mu S$ within 1 to 7 seconds from event onset. SCR corresponds to an increase in $\mu S$ amplitude from through to peak, with increases smaller than 0.02 $\mu S$ given a value of 0.

**Fear potentiated startle.** Fear potentiated startle (FPS) was measured in accordance with the recommendations from the Committee report: Guidelines for human startle eyeblink electromyographic studies [46]. FPS measures the response to the conditioned stimuli by electromyography (EMG) surface measurement of the orbicularis oculi (i.e., muscle located under the lower eyelid that closes the eye during a blink). This measure is used to assess the startle response of the participant. A loud white noise (40 msec; 104dB) was presented at each presentation of the CS through headphones (model AKG K92 closed back studio) to participants to elicit a startle response. EMG was measured throughout the study and was recorded with Bio-Lab Acquisition Software 3.1.13 from MindWare Technologies Ltd. The leads were connected to a 16-Channel Electrode Box, after the skin surface was cleaned with alcohol. To assess the activity of the orbicularis oculi muscle, one electrode (diameter of 5mm Ag/Ag-Cl unshielded electrode filled with Signa Gel) was placed below the lower eyelid on the left side in line with the pupil when looking straight. A second electrode was placed about 1-2cm laterally from the first electrode. A third electrode acted as the isolated ground electrode and was placed on the forehead. The impedance of electrodes was lower than 10 kΩ.

Data were preprocessed in the EMG analysis software (version 3.1.5) from Mindware Technologies Ltd. We segmented data from 0 to 15 seconds post-event onset. We applied a band-pass filter between 28 hz and 500 hz, a notch filter of 60 hz, and a lowpass filter of 5 hz. The data were rectified. All data were visually inspected and rejected when excessive noise distorted the signal, or impedance levels were too high.

## Procedure

Testing took place on three consecutive days each 24 hours apart (see Fig 1 for a detailed depiction of procedure). Participants were connected to all electrodes throughout the study, and

## Study Procedure

**Fig 1. Study procedure.** *Note.* 3-day protocol used for the present study. CS represents 'conditioned stimulus' and US represents 'unconditioned stimulus'. The CS consisted of 2 distinct pictures of spiders and the US was a mild electrical shock. The image that was paired with the shock was counterbalanced across groups. CS+ and CS- presentations were randomized.

SCR and facial EMG were measured throughout the study. On all days, testing sessions began with a 5- minute baseline, during which participants were asked to sit quietly, followed by 10 habituation startle probes (i.e., loud white noise; 40 msec; 104dB) to measure baseline FPS.

**Day one.** Informed consent was obtained, and participants completed all self-report measures. In all conditions, participants were connected to SCR, EMG, and shock electrodes. Following the baseline and startle probe habituation trials, fear acquisition too place. Participants were instructed: *"We are going to start. There will be two images presented. The shock will only be paired with one image. Monitor the relationship between the image you are seeing and when a shock is received. Please keep your eyes on the screen at all times."* Participants underwent fear conditioning, which consisted of eight trials of the CS+ and eight trials of the CS- in pseudo random order. The CS+ was paired with the shock 75% of the time and the CS- was never paired with the shock. Each CS was presented for eight seconds with an inter-trial interval between 10 to 12 seconds during which participants were presented with a black screen with a white cross. The US presentations lasted 200ms. At the end of day one, to establish that the shock contingency was learned, participants were asked which image was paired with the shock. Participants were not invited back for the other two days of testing if they were unable to identify the contingency correctly ($n = 6$).

**Day two.** Participants were randomly assigned to one of three groups: expectation for learning ($n = 16$), no expectation for learning ($n = 16$) or no reactivation ($n = 16$). In all conditions, participants were connected to SCR, EMG, and shock electrodes throughout testing, with the exception that EMG was not measured during reactivation to ensure that the startle probe did not impact reactivation. Baseline and the startle probe habituation trials were completed prior to randomization.

The expectation for learning and no expectation for learning conditions both underwent reactivation which consisted of viewing a single trial of the CS+ in the absence of the US 10 minutes prior to extinction training. Prior to reactivation, these two groups were given separate instructions on the screen and verbally by the experimenter to manipulate the expectation for learning. Participants in the expectation for learning condition were told: *"We are going to start. Shortly you will see the images you saw yesterday. The relationship between the shock and*

*the images has changed. Please observe how it has changed. Please keep your eyes on the screen at all times.*" Participants in the no expectation for learning condition were provided with the following instructions prior to reactivation: *"We are going to start. You will see the same images you saw yesterday. However, today you <u>WILL NEVER</u> receive a shock at any point during the experiment. Please keep your eyes on the screen at all times.*" Participants were provided with these instructions so they knew what to expect throughout testing to minimize the expectation that something new could be learned. Participants in these two groups then had their memory reactivated via a single presentation of the CS+. Participants in these two groups then took a 10-minute break where they watched a TV show. Participants in the no reactivation condition were not exposed to the single presentation of the CS+ (i.e., their condition fear memory was not reactivated) and instead proceed straight to the 10-minute break.

Following the 10-minute break, all participants underwent extinction, consisting of 10 presentations of the CS+ and 11 presentations of the CS- in the absence of the shock. To ensure that exposure to the CS+ and CS- were equivalent across groups on day two, the no reactivation group received one additional CS+ during extinction, since participants in the two reactivation groups had already received one presentation of the CS+ prior to extinction. Before extinction, participants in all three conditions were provided with the following instructions: *"We are going to start. Please monitor the relationship between the image and when a shock is received. Please keep your eyes on the screen at all times. Are you ready?".*

**Day three.** In all conditions, participants were connected to SCR, EMG, and shock electrodes. Following baseline and habituation EMG trials, participants underwent reinstatement, consisting of four unpaired presentations of the US, followed by a 10-minute break during which they viewed another clip of the TV show. Following the break, participants underwent re-extinction, of which the first trial was always the CS+, and the subsequent 9 CS+ trials and 10 CS- trials were presented in pseudorandomized order with half of the participants receiving a CS+ presentation on the second trial and the other half receiving a CS- presentation on the second trial. Participants were then disconnected from the electrodes, debriefed, and compensated for their time.

Prior to debriefing, participants in the expectation for learning and no expectation for learning conditions were administered a manipulation check to determine if the level of expectancy to receive a shock prior to reactivation on day two was successfully manipulated. Participants in the no reactivation condition were not administered the question because the level of expectancy to receive a shock was not manipulated on day two.

## Power analysis

G* power 3.1.9.2 was used to estimate the sample size [47], based on the most complex analysis, a 3 x 2 x 2 mixed factorial ANOVA. A medium effect (*f* = .25) was estimated as research on prediction error have found medium to large effects (e.g., $\eta^2_{p}$ = .10–.21; [32, 35]). Though correlation among the repeated measures (e.g., skin conductance response and fear potentiated startle) has not been reported on in previous studies, we expected physiological measures at Time 1 and Time 2 would be highly correlated, as such, an *r* = .5 was used to assist in calculating power. According to G*power, a sample size of 42 would be needed to achieve a power of .80 with an alpha of .05 and effect size set to f = .25 (medium effect; [48]). Assuming 5% attrition, a total sample of 44 was needed.

## Statistical analysis

Statistical analysis was performed using SPSS software version 23 [49]. SCR and FPS data were skewed, and therefore the data was square root transformed to correct for violations of

normality. Assumptions for homogeneity of variance were checked using the Fmax rule defined as variance discrepancies exceeding 10:1, we note any evidence of discrepancies in the results section.

To facilitate interpretation of results, the early acquisition phase for SCR and FPS was calculated by taking the averages of trials two and three on day one. The first trial for both the CS+ and the CS- were disregarded to help reduce the impact of an orientating effect. The late acquisition phase for both SCR and FPS were calculated by taking the averages of trials seven and eight on day one.

For extinction, the early phase of extinction was calculated by taking the mean of the second and third presentation of the CS+ and CS-. Note that the first presentation of the CS+ for the reactivation groups, actually occurred during reactivation. The late phase of extinction was calculated by taking the averages of the tenth and eleventh presentations of the CS+ and CS- on day two. The late and early phase for extinction were calculated in the same way for SCR and FPS.

Initial analyses were conducted separately to examine if acquisition and extinction were successful. Separate ANOVAs and follow-up planned comparisons were computed for SCR and FPS for acquisition and extinction with Group (expectation for learning vs. no expectation for learning vs. no reactivation) as the between-participant factor, and Stimulus (CS+ vs. CS-), and Time (Phase) as within participant factors.

To test our main analysis of interest, the effect of expectation of learning on reconsolidation, a mixed ANOVA was calculated with Group (no-reactivation vs. reactivation with expectation for learning vs. reactivation with no expectation for learning) as the between-participant factor and Time (last trial of extinction on day 2 vs. first trial of re-extinction on day 3) and Stimulus (CS+ vs. CS-) as the within-participant factors. The main analyses were calculated in the same way for SCR and FPS. It was expected that there would be a Stimulus by Time by Group interaction whereby participants in the reactivation with expectation for learning group would not show a return of fear, whereas the other two groups would exhibit a return of fear as demonstrated by an increase in SCR and FPS from the last trial of extinction to the first trial of re-extinction.

## Results

### Participant characteristics

The final sample consisted of 48 participants with a mean age of 19.28 ($SD$ = 1.86), 62% female and 38% male. A one-way ANOVA showed no differences between the groups with regards to age, $F (2, 44)$ = 1.20, $p$ = .31. A chi-squared revealed no differences with regards to gender, $\chi^2 (3)$ = .67, $p$ = .72. Table 1 provides a summary of participants' scores on the STAI-T, STAI-S, and the SPQ, as well as the mean voltage selected by participants. One-way

**Table 1. Participants means on the anxiety and spider fear measures.**

| Variable | Expectation for learning group M (SD) | No expectation for learning group M (SD) | No reactivation M (SD) |
|---|---|---|---|
| Voltage | 37.66 (8.58) | 34.16 (10.12) | 41.09 (9.32) |
| SPQ | 9.05 (5.63) | 8.12 (6.36) | 7.80 (8.06) |
| STAI-T | 41.21 (9.83) | 41.82 (9.03) | 40.80 (11.03) |
| STAI-S | 40.53 (9.27) | 48.04 (10.96) | 42.53 (53) |

*Note.* M; Mean; SD; Standard deviation; SPQ; Spider Phobia Questionnaire [40]; STAI-T; Spielberger Trait Anxiety Inventory [42]; STAI-S; Spielberger State Anxiety Inventory [42].

ANOVAs found no differences between groups on the voltage selected, $F(2, 44) = 2.12$, $p = 13$, the STAIT-T, $F(2, 44) = .04$, $p = .96$, the STAI-S, $F(2, 44) = 2.29$, $p = .11$, or the SPQ, $F(2, 44) = .14$, $p = .87$.

## Manipulation check

An independent t-test was computed to determine if there were differences in ratings taken at the end of the study between the expectation for learning and no expectation for learning conditions with regards to how much they expected to receive a shock on day two. We predicted that participants in the expectation for learning condition would report a greater expectation to receive shock on day two. Participants in the expectation for learning condition did not differ ($M = 4$, $SD = 1.48$, min = 0 and max = 5), from the no expectation for learning condition ($M = 3.81$, $SD = 1.38$, min = 0 and max = 5), $t(25) = .34$, $p = .74$, d = .13, with regards to the degree to which they expected to receive a shock on day two.

## Skin conductance response

**Acquisition.** To establish that participants across the three groups underwent successful acquisition on day one, we conducted a Stimulus (CS+ vs. CS-) x Time (Day 1: early vs. late) x Group (expectation for learning vs. no expectation for learning vs. no reactivation) mixed ANOVA. Fig 2 displays the mean of each trial during acquisition. We found main effects for Stimulus, $F(1, 43) = 6.44$, $p = .02$, $\eta^2_p = .13$, and Time, $F(1, 43) = 15.18$, $p = < .001$, $\eta^2_p = .26$, but no effect for Group, $F(2, 43) = 1.27$, $p = .29$, $\eta^2_p = .06$. As expected, the main effects were moderated by a Stimulus x Time interaction, $F(1, 43) = 6.58$, $p = .01$, $\eta^2_p = .13$. There were no other two- or three-way interactions, $Fs(1, 43) < .74$, $ps > .48$, $\eta^2_p < .03$. As seen in Fig 2, there was no difference in the CS+ versus CS- at the start of acquisition, $t(45) = —.31$, $p = .76$, $d = .05$, but by the end of acquisition participants had higher SCRs to the CS+ than the CS-, $t(45) = 3.13$, $p = .003$, $d = .48$.

**Extinction.** To assess if participants underwent successful extinction on day two, we conducted a Stimulus (CS+ vs. CS-) x Time (Day 2: early vs. late) x Group (expectation for learning vs. no expectation for learning vs. no reactivation) mixed ANOVA. Fig 2 displays the mean of each trial during extinction. As expected, we found main effects of Stimulus, $F(1, 43) = 4.756$, $p = .03$, $\eta^2_p = .10$, and Time, $F(1, 43) = 26.93$, $p = < .001$, $\eta^2_p = .39$ but no main effect for Group $F(2, 43) = .61$, $p = .54$, $\eta^2_p = .03$. None of the two-way or the three-way interactions were meaningful, $Fs(1, 43) < 1.03$, $ps > .37$, $\eta^2_p < .05$. Given our a priori hypothesis (i.e., we predicted that participants in all groups would have a greater SCR to the CS+ than the CS- at the start of extinction and no difference in their SCR to the CS+ and CS- at the end), follow-up t-tests were calculated to compare the mean SCR to the CS+ and the CS- at the start of extinction and at the end of extinction. As seen in Fig 2, participants had a greater SCR to the CS+ than the CS- at the start of extinction, $t(45) = 3.38$, $p < .001$, $d = .50$, whereas there were no differences in SCR to the CS+ and CS- at the end of extinction, $t(45) = .48$, $p = .63$, $d = .07$.

**Reconsolidation.** To examine whether the expectation for learning prior to memory reactivation prevents reinstatement of the conditioned fear response, a Stimulus (CS+ vs. CS-) x Time (Last Trial of Extinction on Day 2 vs. First Trial of Re-extinction on Day 3) x Group (expectation for learning vs. no expectation for learning vs. no reactivation) mixed ANOVA was conducted. Fmax was 11.5:1 between the no expectation for learning group and the expectation for learning group, and 9.5:1 between the no reactivation group and the expectation for learning group, suggesting that there was heterogeneity of variance. There were no main effects for Stimulus, $F(1, 43) = .03$, $p = .87$, $\eta^2_p < .01$, or Group, $F(2, 43) = .44$, $p = .65$, $\eta^2_p = .02$, but there was a main effect of Time, $F(1, 43) = 42.18$, $p < .001$, $\eta^2_p = .50$. Contrary to

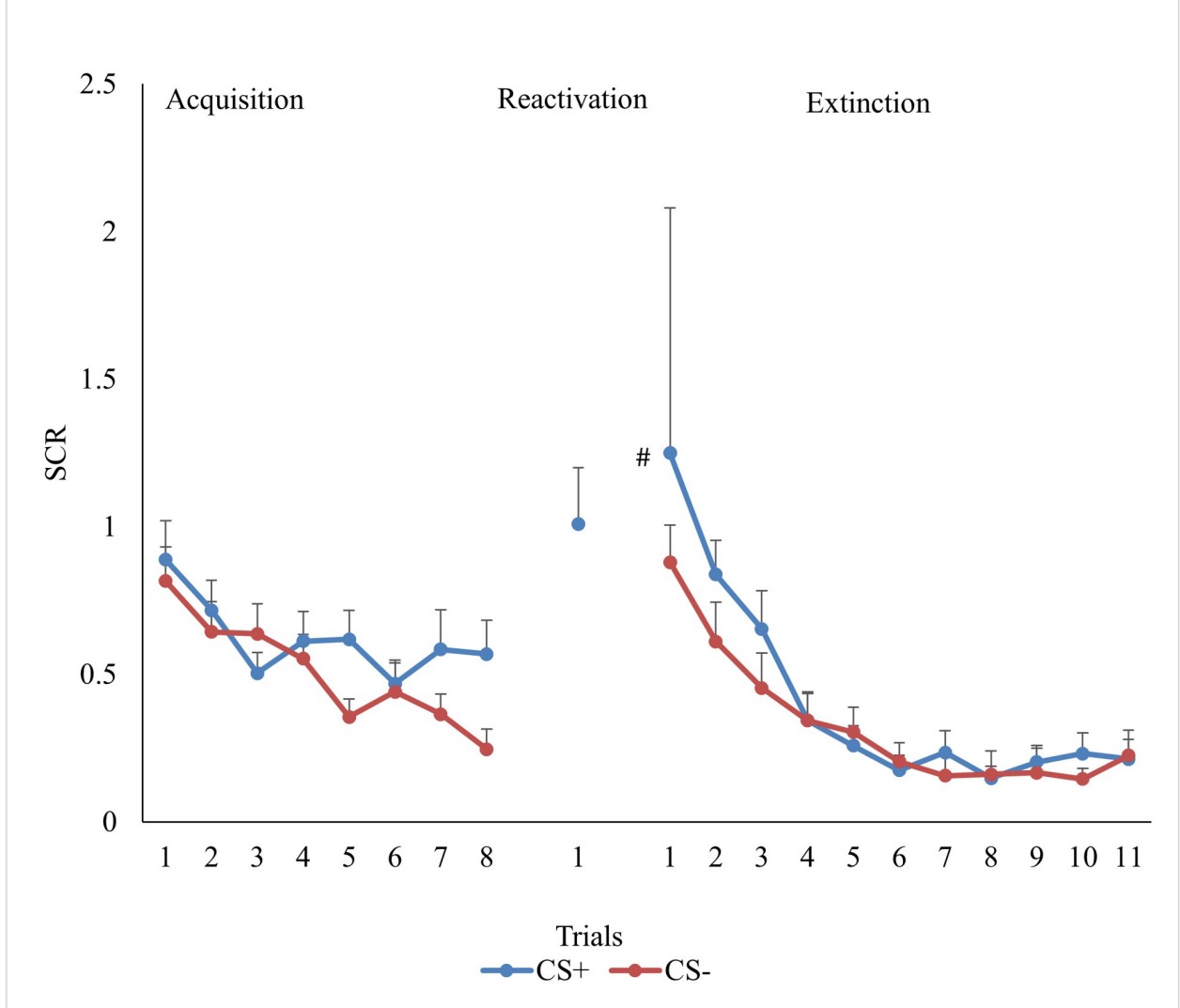

**Fig 2. Mean SCR trial results for acquisition and extinction.** *Note.* Mean skin conductance response (SCR) across trials collapsed across groups. Acquisition consisted of presentations of the CSa+ that was sometimes paired with the US and the CS- that was never paired with the US. Extinction consisted of the presentation of the CSa+ and the CS- without the US. Measures of SCR were taken at every presentation of a stimuli. * EDA for groups that received reactivation. # EDA for the first trial of the CS+ for the group that did not receive reactivation.

predictions, there were no two-way or the three-way interactions, $Fs$ $(1, 43) < 1.39$, $ps > .26$, $\eta^2_p < .04$. However, given the main effect of time and our a priori hypothesis, separate follow-up t-tests were run to compare the return of fear in each of the groups (i.e., The last trial of extinction on day two compared to the first trial of re-extinction on day three). As seen in Fig 3, inconsistent with our predictions, SCR increased for both the CS+ and CS- among participants in the expectation for learning condition, CS+, $t(14) = -3.27$, $p = .01$, $d = -.89$; CS-, $t(14) = -3.82$, $p = 002$, $d = -1.32$, and the no expectation for learning condition, CS+, $t(14) = -2.56$, $p = .02$, $d = -.91$; CS-, $t(14) = -4.13$ $p = .001$, $d = -.90$. The no reactivation group showed an increase in their SCR to the CS+, $t(15) = -2.47$, $p = .03$, $d = -.84$, but not to the CS-, $t(15) = -1.86$, $p = .08$, $d = -.58$, from the end of extinction on day two to the beginning of re-extinction on day three. Thus, participants, regardless of condition, showed a return of fear on day three. That is, we observed no evidence that reconsolidation took place, as measured by SCR.

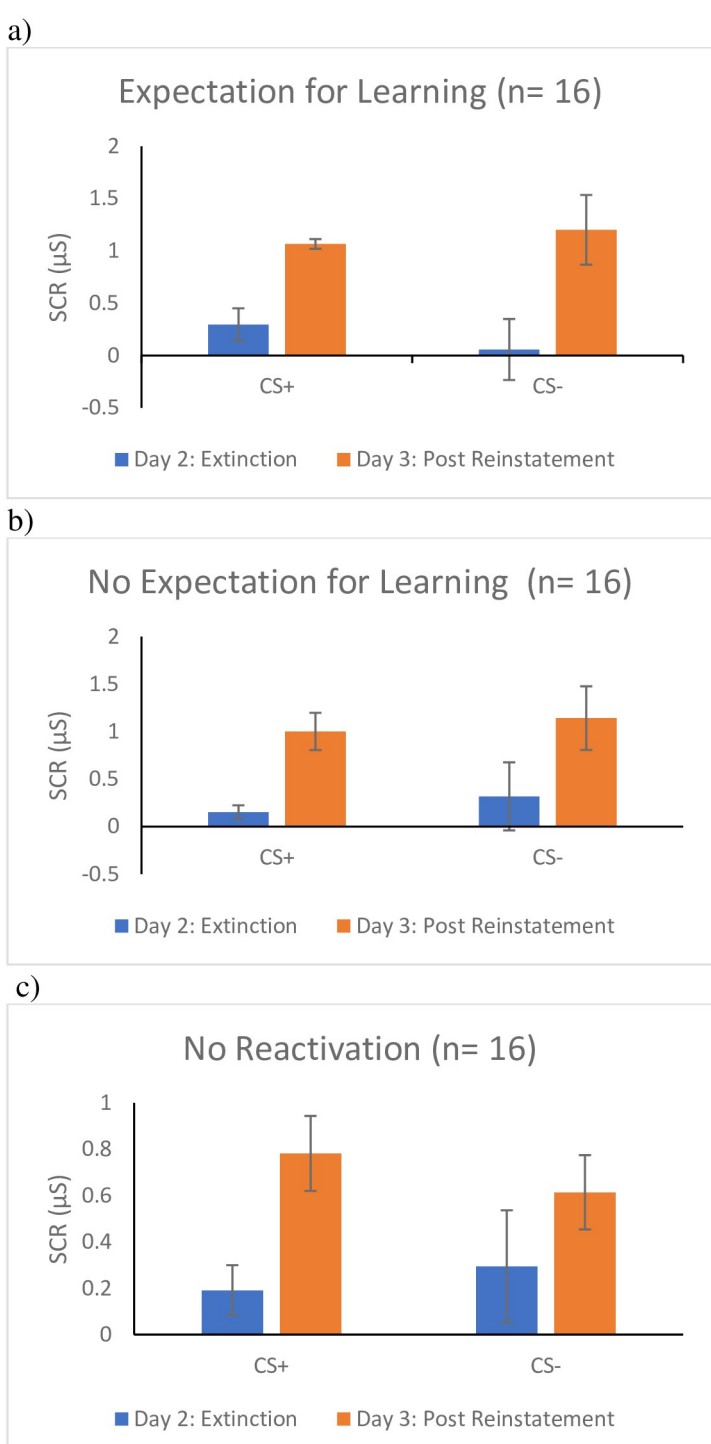

**Fig 3. Return of fear following reinstatement as measured by SCR for the a) Expectation for learning group, b) the no Expectation for learning group, and c) the No reactivation group.** *Note.* Results for skin conductance (SCR) for reconsolidation. Day 2 is a measure of fear following extinction (i.e., last trial of extinction). Day 3 is a measure of fear following reinstatement of fear (i.e., first trial of re-extinction). Measures of SCR were taken at every presentation of a stimuli.

### Fear potentiated startle

The same statistical analyses outlined above for SCR were computed for FPS.

**Acquisition.** Fig 4 depicts participants' mean FPS response to each trial during acquisition. We found a main effect for Time, $F(1, 45) = 25.77$, $p = < .001$, $\eta^2_p = .36$, and Stimulus, $F(1, 45) = 5.56$, $p = .02$, $\eta^2_p = .11$, but no main effect for Group, $F(2, 45) = 4.05$, $p = .02$, $\eta^2_p = .15$. None of the two-way interactions were meaningful, $Fs(2, 45) < 2.59$, $ps > .09$, $\eta^2_p < .10$, however a Stimulus x Time x Group interaction was detected, $F(2, 45) = 4.05$, $p = .02$, $\eta^2_p = .15$.

Follow up analysis were run given the three-way interaction. For the expectation for learning group, there were no differences in the CS+ compared to the CS- at the start,

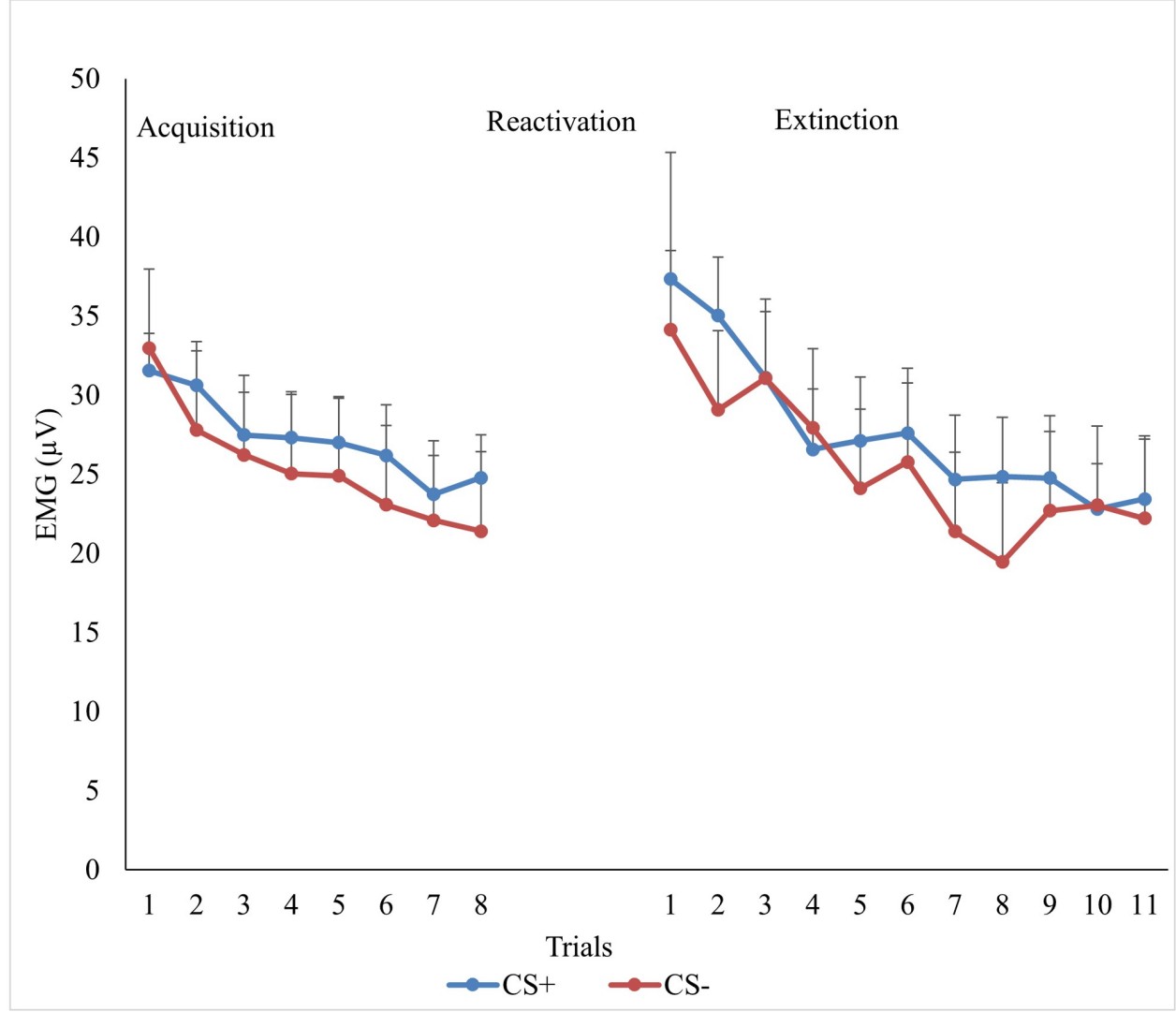

**Fig 4. Mean FPS trial results for acquisition and extinction.** *Note.* Mean fear potentiated startle (FPS) across trials collapsed across groups. Acquisition consisted of presentations of the CSa+ that was sometimes paired with the US and the CS- that was never paired with the US. FPS was not measured at reactivation so that fear response was not influenced by startle probe. Extinction consisted of the presentation of the CSa+ and the CS- without the US. The error bars represent standard error. Measures of FPS were taken at every presentation of a stimuli. # Note that EMG for the first CS+ trial is for the no reactivation group only. EMG was not measured during the reactivation trial for the other two groups as we did not want the startle probe to interfere with reconsolidation.

$t$ (16) = .26, $p$ = .80, $d$ = .03, or the end of acquisition, $t$ (15) = .72, $p$ = .48, $d$ = .11. For the no expectation for learning group, there was no difference in FPS to the CS+ compared to the CS- at the start of acquisition, $t$ (15) = .49, $p$ = .63, $d$ = .06, but FPS was greater for CS+ than the CS- at the end of acquisition, $t$ (15) = 2.37, $p$ = .03, $d$ = .4. For the no reactivation group, participants showed greater FPS to the CS+ than the CS-, $t$ (15) = 2.65, $p$ = .02, $d$ = .33, at the start of acquisition, but did not by the end, $t$ (15) = -.19, $p$ = .85, $d$ = -.03. Thus, only the no expectation for learning condition displayed fear acquisition based upon their FPS, though as noted in our method section, all participants could successfully identify the contingency between the shock and CSs.

**Extinction.** Fig 4 shows participants' mean FPS to each trial during extinction. As expected, we found main effects of Stimulus, $F$ (1, 45) = 5.26, $p$ = .03, $\eta^2_p$ = .11, and Time, $F$ (1, 45) = 33.61, $p$ = < .001, $\eta^2_p$ = .43, but no main effect for Group, $F$ (2, 45) = .001, $p$ = .99, $\eta^2_p$ = < .001. None of the other two-way or the three-way interactions were meaningful, $Fs$ (1, 45) < 1.84, $p$s > .17, $\eta^2_p$ < .08. We calculated follow-up $t$-tests comparing the mean FPS to the CS + and the CS- at the start of extinction and at the end of extinction. Consistent with our expectations, participants had a larger FPS to the CS+ than the CS- at the start of extinction, $t$ (47) = 2.13, $p$ = .04, $d$ = .16, but by the end of extinction there was no difference between the CS+ and CS-, $t$ (45) = 1.09, $p$ = .28, $d$ = .07. These results suggest that extinction occurred in all groups, but the magnitude of this effect was not large.

**Reconsolidation.** Our 2 (Stimulus) x 2 (Time) x 3 (Group) ANOVA showed no main effects for Stimulus, $F$ (1, 45) = 2.65, $p$ = .11, $\eta^2_p$ = .06, or Group, $F$ (2, 45) = .22, $p$ = .80, $\eta^2_p$ = .01, but there was a main effect of Time, $F$ (1, 45) = 14.25, $p$ < .001, $\eta^2_p$ = .24. There was also a Stimulus x Group interaction, $F$ (2, 45) = 3.57, $p$ = .04, $\eta^2_p$ = .14. Inconsistent with our predictions, no other two-way or the three-way interactions were meaningful, $Fs$ (1, 45) < 1.60, $p$s >.21, $\eta^2_p$ < .07. Given the main effect of Time, the two-way interaction of Stimulus by Group, and our a priori hypothesis, follow-up t-tests were calculated. As seen in Fig 5, participants in the expectation for learning group showed no difference in their FPS to the CS+, $t$ (15) = -1.20, $p$ = .25, $d$ = -.29 or CS-, $t$ (15) = -.93, $p$ = .37, $d$ = -.25, from the last trial of extinction on day two to the first trial of re-extinction on day three, suggesting that there was no return of fear. In the no expectation for learning condition participants showed an increase in FPS to the CS+, $t$ (15) = -2.72, $p$ = .02, $d$ = -.75, and the CS-, $t$ (15) = -3.27 $p$ = .01, $d$ = -.99, from the end of extinction of day two to the beginning of re-extinction following reinstatement on day three, suggesting that they did experience a return of fear. The no reactivation group (i.e., extinction alone) also showed an increase in FPS from the end of extinction on day two to the beginning of re-extinction on day three to the CS+, $t$ (15) = -3.76, $p$ = .01, $d$ = -.54, but not the CS-, $t$ (15) = -1.59, $p$ = .13, $d$ = -.28, again suggesting that they also experienced a return of fear.

**Post hoc analysis.** Given that not all groups displayed physiological fear acquisition with FPS as the measure of fear, we ran a post hoc analysis including only participants that displayed greater FPS to the CS+ than CS- during the late phase acquisition (n = 20). Though clearly underpowered, we report paired $t$-tests comparing FPS at the end of day 2 to the start of day 3 for the CS+ and CS-. The pattern of response was similar to that found in the entire sample; FPS response remained stable in the expectation for learning to the CS+, $t$ (4) = -.06, $p$ = .96, $d$ = -.03, and CS-, $t$ (4) = .18, $p$ = .87, $d$ = .08. FPS also remained stable in the no expectation for learning group for the CS+, $t$ (5) = -1.26, $p$ = .27, $d$ = -.51, and CS-, $t$ (5) = -1.18, $p$ = .29, $d$ = -.48. Finally, in the no reactivation group FPS increased from Day 2 to Day 3 for the CS+, $t$ (8) = -3.40, $p$ = .009, $d$ = -1.14, but not for the CS-, $t$ (8) = -.73, $p$ = .49, $d$ = -.24 in the no reactivation group.

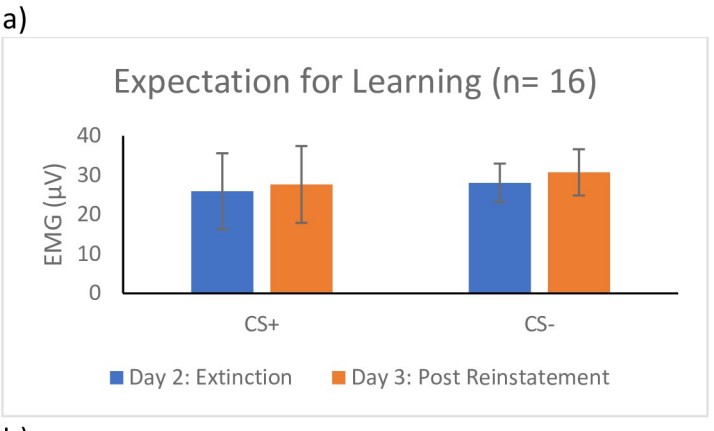

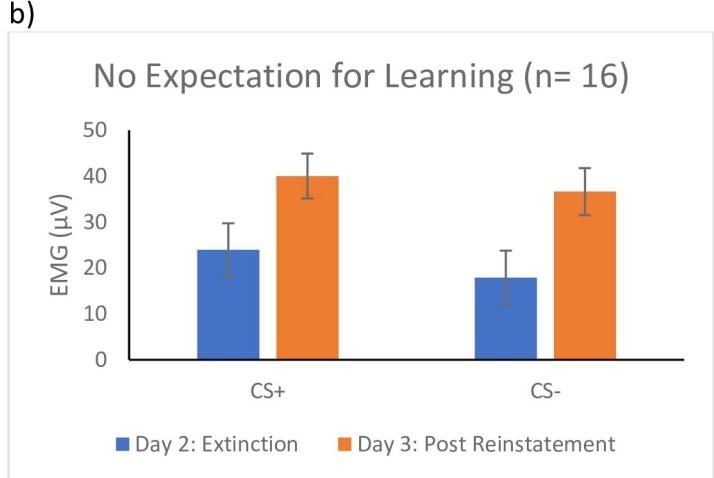

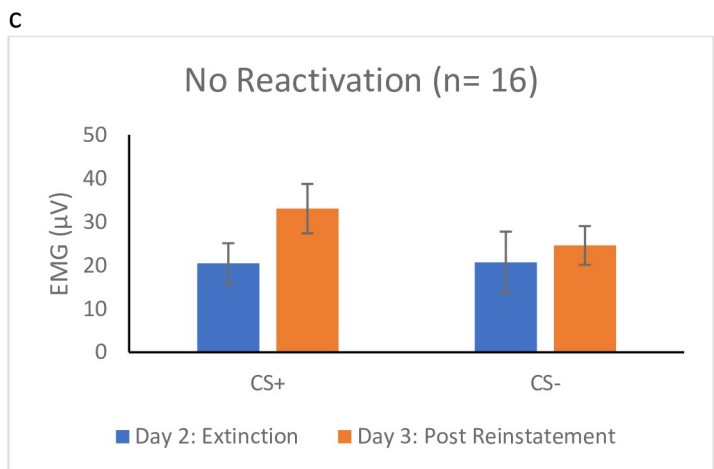

**Fig 5. Return of fear following reinstatement as measured by FPS for the a) Expectation for learning group, b) the no Expectation for learning group, and c) the No reactivation group.** *Note.* Results for fear potentiated startle (FPS). Day 2 is a measure of fear following extinction (i.e., last trial of extinction). Day 3 is a measure of fear following reinstatement of fear (i.e., first trial of re-extinction). The error bars represent standard error. Measures of FPS were taken at every presentation of a stimuli.

## Discussion

The aim of the current study was to examine if the expectation for learning impacts the reconsolidation of conditioned fear memories using the post-retrieval extinction paradigm in an undergraduate sample using both SCR and FPS as measures of the fear response. The study found a different pattern of results depending on whether SCR or FPS was the unit of measurement. Specifically, for SCR, we found participants across all three conditions (i.e., the expectation for learning, the no expectation for learning, and the no reactivation) exhibited a return of fear following reinstatement on day three of testing. In contrast, FPS remained stable in the expectation for learning group, whereas the no expectation for learning and no reactivation groups showed an increase in FPS to the CS+, suggesting there was a return of fear in the latter two groups. However, this needs to be interpreted in light of the fact that there was no evidence of fear acquisition as measured by FPS, and that the effect size for change in FPS from the end of extinction to start of reextinction in the expectation for learning group was small. Indeed, when looking within only participants that showed evidence of acquisition, FPS remained stable in both reactivation groups, but there was evidence of a return of fear in the no reactivation group. The expectation for learning did not appear to enhance reconsolidation when fear was measured via FPS or SCR.

Our results are inconsistent with previous studies that have found that enhancing learning expectancy or inducing a prediction error enhanced reconsolidation [32, 35–38]. One potential explanation, and consistent with our failed manipulation check, is that our verbal manipulation of expectancy for learning was not salient enough to differentiate the groups. In addition to verbal instructions, Sevenster et al. [35] maximized differences across groups by not connecting participants in the no expectation for learning group to the shock equipment to ensure that they absolutely could not expect to learn something new during reactivation. With the rationale of keeping groups similar in all respects, except for the expectation for learning, we chose to connect participants in the no expectation for learning group to the shock equipment and only provided verbal instructions that they would not receive a shock. It is possible that manipulating the expectation for learning needs to be more salient, and this could explain differences across these two studies. In support of this suggestion, Yang et al. [50] found that strong prediction errors were more likely to induce reconsolidation compared to weaker prediction errors.

Future research should further explore the possible association between the strength of the prediction error or learning expectancy and reconsolidation. Indeed, in contrast to our findings, previous suggests that verbal contingency instructions can have a marked effect on fear conditioning [51] and extinction [30]. Mertens and colleagues [51] found that precise and general verbal contingency instructions both lead to stronger differential conditioning for FPS and SCR compared to receiving no instructions. However, the authors noted that it is not evident how verbal contingency instructions affect extinction or return of fear. We would add that future research should also examine if different contingency instructions influence reconsolidation.

Another notable finding is that different patterns of learning, both with regards to acquisition and reconsolidation, were observed between SCR and FPS. Though SCR and FPS are both frequently used in fear conditioning research either separately or together [52–55], there is growing consensus that they measure different constructs. Specifically, SCR is conceptualized as a measure of general arousal and attention towards salient information, whereas FPS appears to be more a direct measure of fear or anxiety [54, 56, 57]. That we only found evidence of acquisition on SCR is perhaps not surprising since we only included participants who demonstrated contingency awareness. That acquisition was not as apparent for FPS, suggests

that fear of the CS+ was not as easily acquired, and only clearly acquired for a subgroup of participants. Soeter and Kindt [57] found that propranolol disrupted reconsolidation as measured by FPS, but not as measured by SCR. They suggest that as propranolol specifically acts on β-adrenergic receptors in the amygdala [58], it is unsurprising the physiological measures more directly linked to fear showed evidence of reconsolidation disruption, while those associated with contingency awareness did not. At least two other studies using the post-retrieval extinction design have only found evidence of reconsolidation for measures of FPS but not SCR [4, 35], though see Golkar et al. [25] was able to find evidence of reconsolidation using both measures at the same time. Researchers studying reconsolidation should be careful in how they employ inclusion and exclusion criteria. We used contingency awareness as our criteria for inclusion, whereas other studies have used differences in SCR or FPS between the CS+ and CS- at the end of acquisition [e.g., 1]. If we had defined inclusion based upon other factors, such as differential FPS response to the CS+ or increased self-reported fear to the CS+, it is possible results may have differed. This recommendation is consistent with a recent study demonstrating that the way in exclusion criteria were employed in the Schiller et al. [1] paper influenced their findings [59]. Furthermore, use of fear conditioning designs that maximizes evidence of fear conditioning (e.g., using 100% reinforcement rates for the CS+, [60]) may also be beneficial when designing studies aimed examining reconsolidation of emotional memories using the post-retrieval extinction paradigm.

Although memory reconsolidation has the potential to be used to enhance the treatment of clinical disorders maintained by maladaptive memories (e.g., posttraumatic stress disorder, phobias, anxiety disorders), this study highlights the need for a better understanding of reconsolidation-based behavioural interventions prior to clinical application. Though including strategies to enhance learning expectancy prior to extinction may be beneficial based upon findings from previous pharmacological studies [32, 35–38], results from the current study highlight that such strategies likely need to be quite salient, and that simple verbal manipulations may not be sufficient.

## Limitations and future research

There are several limitations presented above that should be taken into account when considering these results and highlights the need for further research. First, as previously highlighted, participants did not exhibit successful physiological fear acquisition on day one with FPS as the measure of fear, as defined by greater FPS to the CS+ than the CS-. However, it should be noted that differences between the CS+ and CS- were in the expected direction, albeit small. Although some previous studies have excluded participants that did not demonstrate successful fear acquisition [e.g., 1, 4, 25, 61], research suggests that a lack of physiological fear acquisition does not predict learning during extinction [62, 63]. Furthermore, Lonsdorf et al. [63] suggests that excluding participants that do not demonstrate successful physiological fear acquisition may create a sample of exceptional learners that are not representative of the population. As such, the lack of physiological fear acquisition in the present study may demonstrate a more accurate representation of fear conditioning and reconsolidation in the general population. Future research should further explore whether the ability to update reconsolidation varies depending on the type of memory that's measured (e.g., emotional memory versus declarative memory/contingency learning) and also using different measurement strategies (e.g., FPS, SCR, behavioural, self-report).

Furthermore, these results need to be interpreted in light of the fact that no differences were observed in ratings between the expectation for learning and the no expectation for

learning groups with regards to the expectation of receiving a shock on day two. Thus, it is possible that we did not successfully manipulate the expectancy to learn something new during reactivation. However, upon reflection, the question asked may not have accurately measured participants' expectancy to learn something new but simply measured whether they expected to receive a shock during reactivation. Additionally, as the question was asked on day three (to ensure that the manipulation check did not interference with the actual experiment), it is possible that participants' responses might not accurately represent their expectation to receive a shock on day two. Finally, this question was not asked to participants in the no reactivation group. Therefore, we are unable to determine whether this group expected to receive a shock on day 2.

Next, for SCR, the assumption of homogeneity of variance was violated for the main analysis. Though this may have reduced confidence in the findings from the mixed ANOVA, follow-up tests used pair-wise comparisons within groups, and therefore should not be affected by this violation.

Lastly, although an a priori power analysis indicated that the present study was sufficiently powered for a medium effect, several of the interactions in the present study had *p*-values above .05 but had effect sizes in the medium range. Thus, it is possible that a Type II error might account for the *p*-values above .05. Given the medium effects and the limited as well as inconsistent research on reconsolidation, it could be beneficial to replicate the studies with a larger sample size to minimize a Type II error and to better understand factors that account for the large variability among participants (e.g., level of acquisition or extinction, conditioned memory versus natural memories), so that these can be better controlled in future research. It is also possible that effects using behavioural methods may yield small to medium effects, and thus require larger sample sizes to detect the effect. Smaller effects observed in behavioural studies may help explain some of the replication issues. Thus, replicating these studies with larger samples and reporting on effect sizes may help understand who is likely to benefit from reconsolidation related interventions.

## Conclusions

Despite these limitations, the findings from the current study have important implications for the study of reconsolidation and the study of fear learning and memory more generally. Our results suggest that a verbal manipulation of the expectation for learning does not enhance reconsolidation. Future research should consider exploring how strong instructions need to be to induce expectancy violations. Though we did find some evidence that reconsolidation occurred for the fear memory, as measured by FPS, this evidence was weak and only apparent in the subgroup of participants demonstrating fear acquisition for FPS. Differences in fear acquisition between FPS and SCR may be explained by the fact that acquisition was defined by contingency awareness, which is more closely linked to SCR than to FPS. Overall, the findings highlight how sensitive results from fear learning research are to research design decisions and support a growing consensus that research studying fear learning processes, including reconsolidation, must attend carefully to research design choices, including how to define learning and the type of outcome measures selected.

## Supporting information

**S1 Data.**
(DOCX)

## Acknowledgments

We wish to acknowledge the following University of Ottawa undergraduate students who assisted in data collection for this study: Amanda Dewsbury, Ila Lennips, and Hannah Zang.

## Author Contributions

**Conceptualization:** Julia Marinos, Andrea R. Ashbaugh.

**Data curation:** Olivia Simioni.

**Formal analysis:** Julia Marinos, Olivia Simioni.

**Funding acquisition:** Andrea R. Ashbaugh.

**Methodology:** Julia Marinos.

**Project administration:** Julia Marinos, Andrea R. Ashbaugh.

**Resources:** Andrea R. Ashbaugh.

**Supervision:** Andrea R. Ashbaugh.

**Writing – original draft:** Julia Marinos.

**Writing – review & editing:** Olivia Simioni, Andrea R. Ashbaugh.

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
