## [Decision Letter · Decision Letter 0]

29 Jan 2021

PONE-D-20-33271

Impact of the expectation on memory reconsolidation using a post retrieval extinction paradigm

PLOS ONE

Dear Dr. Ashbaugh,

Thank you for submitting your manuscript to PLOS ONE. After careful consideration, we feel that it has merit but does not fully meet PLOS ONE’s publication criteria as it currently stands. Therefore, we invite you to submit a revised version of the manuscript that addresses the points raised during the review process.

We look forward to receiving your revised manuscript.

Kind regards,

Alexandra Kavushansky, PhD

Academic Editor

PLOS ONE

Journal Requirements:

Reviewers' comments:

Reviewer's Responses to Questions

**Comments to the Author**

1. Is the manuscript technically sound, and do the data support the conclusions?

Reviewer #1: No

Reviewer #2: No

2. Has the statistical analysis been performed appropriately and rigorously? 

Reviewer #1: Yes

Reviewer #2: Yes

3. Have the authors made all data underlying the findings in their manuscript fully available?

Reviewer #1: No

Reviewer #2: No

4. Is the manuscript presented in an intelligible fashion and written in standard English?

Reviewer #1: No

Reviewer #2: Yes

5. Review Comments to the Author

Reviewer #1: The current research investigated whether expecting to learn something during exposure enhances reconsolidation of fear memory. While this research question may be relevant to understand the boundary conditions of reconsolidation, several conceptual and methodological issues arise that limit interpretation of the current findings. Most importantly, the failed manipulation and fear acquisition, and unclear statements in the introduction and discussion section greatly devalue the manuscript.

Title:

- Unclear what is meant by ‘the expectation’

Abstract:

- The objective is unclear. What do the authors mean with ‘expectation for learning’?

- The study took place ‘on’ three consecutive days.

- The explanation of the manipulation is unclear for two reasons. First, what are the oral instructions? Second, it is hard to follow since memory reactivation was not mentioned yet.

- The results start with the interpretation of the findings. I would recommend merely reporting the findings at this point.

- No keywords are reported.

Introduction:

- Two sentences barely form a paragraph (first paragraph). Please integrate this with the following paragraph.

- Eliminate the word ‘consistently’, because in this sentence it is clear that these are not consistent findings:

“Studies in humans have consistently demonstrated that a conditioned fear response, as

measured by fear potentiated startle (FPS), can be eliminated if reactivation is paired with oral

administration of propranolol [4, 7, 8, 9], though a few studies have failed to replicate this effect

[10, 11].”

- In the following sentence the authors refer to ‘these studies’, yet they only discuss one study in that paragraph. This is somewhat confusing:

“These studies illustrate that reactivated memories following the administration of propranolol, can be modified and this process might be beneficial in the treatment of anxiety based psychological disorders.”

- Why does propranolol only indirectly target the mechanisms involved in reconsolidation? Please explain.

- The authors only briefly touch upon the issues in reconsolidation research. They merely state that inconsistencies in replicating these findings suggest that memory recall may not be sufficient. However, there is already a large body of potential boundary issues that are not discussed. Also, in the work by Schiller, many participants were removed prior to data analyses. This could also explain failed replications later on, but this is not discussed by the authors.

Methods:

- For the power analysis, it is unclear what tests the power analysis was based on. Moreover, is a medium effect the effect that the authors expect? On which studies was that effect size based as there are so many failed replications already?

- What were the criteria for usable physiological data? Later on, the authors state that 12 participants were excluded because their physiological data was not readable. It remains unclear what the exclusion criteria were.

- Why did 33 participants drop out? That was one third of your sample.

- What was the reinforcement rate? ‘Sometimes paired with a shock’ is vague.

- Inconsistency in order when describing STAI-T and STAI-S.

- Why was the manipulation check retrospectively measured? The answers may be influenced by what actually happened.

- The authors pose that SCR is a measure of anxiety. Yet, some authors claim that fear and anxiety are different constructs. In addition, some researchers state that SCR measures arousal and not fear per se. Can the authors please elaborate on that?

- EMG abbreviation is not written in full.

- The authors make a strong statement about what FPS represents:

“This measure is used to assess the startle response of the participant as neurologically it represents the connections from the amygdala to the startle-reflex pathway in the brainstem [35].”

What do the authors mean by this?

- Participants were connected to the skin conductance � incorrect language, please rephrase.

- What was the duration of the CS presentation, US presentation, and intertrial intervals?

- Terminology is not used, such as contingencies and reinforcement rate.

- Are there no noise alone trials measured for FPS?

- “This break allows for the activation of the neural mechanisms needed for reconsolidation to take place [1].” � Unclear what the authors mean by this.

- It is only indirectly stated what the memory reactivation was, namely: “Participants in the No

Reactivation condition were not exposed to the single presentation of the CS+ (i.e., their

condition fear memory was not reactivated) and instead proceed straight to the 10-minute break.”

Please make this more explicit.

- How many extinction trials were there?

- What was the order of trials in the re-extinction (or reinstatement) phase? Which CS was presented first? And was the order counterbalanced across participants?

- Why did participants in the no reactivation group did not receive the manipulation check? Obviously it matters whether the other two groups differs from each other, but also whether they differ from the control group. That would give more direct information about the necessity for prediction error.

- Statistical assumptions were violated, but only the correction for normality violation is reported here. What were the other violations and how is that corrected?

- A lot of data is disregarded by the statistical approach (e.g., first and late acquisition phase). That is quite wasteful. Why was this approach chosen?

Results

- A conclusion from the manipulation check is lacking, but if I understood correctly, the manipulation failed. The No Expectation for Learning condition reported that they still expected to receive a shock and this expectation did not differ from the Expectation for Learning group. Thus, the study cannot answer the research question.

- Did the authors not expect a difference at the first extinction trial? As the intervention was already before the extinction phase it makes sense that the No Expectation for Learning condition displayed reduced conditioned responding compared to the other groups on the first trial in this phase. Yet, since the manipulation was not successful, obviously there were no group differences.

- In addition, results cannot be interpreted since fear acquisition on FPS was unsuccessful (except in the No Expectation for Learning group. Why did the authors analyze the extinction and reconsolidation phase if there was no learning in the first place?

Discussion

- The authors conclude that only in the Expectation for Learning group CS+ responding remained stable following reinstatement. Yet, this is not the case when only the participants with successful fear acquisition were included, since also in the No Expectation for Learning group FPS remained stable. Thus, this conclusion is incorrect. Although the authors state that these findings or not robust and that this effect disappeared when only including participants with successful fear acquisition, the conclusion should be that in two groups CS+ responding remained stable.

- The authors start explaining the dissociation in their data. First of all, there seems to be no dissociation as they did not find any results on both SCR and FPS. Second, with this paradigm, the interventions target reconsolidation of threat memories, not of SCR itself. As a result of successful reconsolidation, SCR responding should be lower. Therefore, the following sentence seems incorrect: “Researchers have suggested that SCR reflects a cognitive representation of arousal and is more difficult to reconsolidate”. Finally, as stated before, the dissociation (or absence of dissociation) between different physiological responses is highly debated. Stating that SCR reflects a cognitive representation of arousal is unclear and seems incorrect.

- The paragraph on loud tones interfering with SCR does not make sense. Obviously loud sounds can interfere with SCR, but there is a plethora of studies demonstrating that FPS and SCR can simultaneously be measured. Moreover, it remains unclear how this would explain an absence of effects, since the expected findings were also absent on FPS.

- The limitations of physiological responding are not properly discussed. How would the authors explain that there was differential responding in fear acquisition, but that it just was not visible until day 2?

- Why is the failed manipulation check only discussed so late in the discussion section? This is a major problem with this study. The authors claim that the manipulation check may not have measured whether the participants expected to learn something. Even if that is the case, it can be argued that participants in both the Expectation for Learning and No Expectation for Learning were learning something (namely that they did not receive shocks, although they expected these). The same holds for the No Reactivation group. Is there any reason to believe that they had different expectations about whether they would learn something?

- There is quite some literature suggesting that verbal responses are quite strong to change expectancies (e.g., Mertens et al., 2020), and actually change fear responding.

APA:

- Schiller et al., [1]. The comma should be omitted.

- Spelling mistakes, such as “i.e., Post-reterival extinction paradigam”, “propanrolol”

- Multiple double spaces

- Language: the manuscript is not densely written and feels quite lengthy.

- Biased language, for instance “62% female”

- Inconsistent use of capital letters, for instance for group names.

Refererences

Mertens, G., Boddez, Y., Krypotos, A. M., & Engelhard, I. M. (2020). Human fear conditioning is moderated by stimulus contingency instructions. Biological Psychology. https://doi.org/https://doi.org/10.1016/j.biopsycho.2020.107994

Reviewer #2: The authors used a retrieval-extinction paradigm to ask the role of expectation in reconsolidation. They recruited 3 groups of 48 undergraduate students who were presented with a spider image that was paired with a shock and another image was not paired (i.e. differential fear conditioning). They were randomly assigned to group 1: reactivation with expectation for learning, group 2: reactivation without expectation, or group 3: no reactivation before they all received an extinction session. On the next day, they received unpaired shocks and further extinctions trials. They found no difference in expectation in the 2 reactivation groups. Based on the SCR measurement, groups 1 and 2 both showed fear return for both CS+ and CS- after reinstatement while group 3 showed fear return for CS+. Based on the FPS measurement, groups 1 and 2 both showed fear return for both CS+ and CS- after reinstatement.

Although this study addresses a subject of importance, it did not cite reference already addressing this specific topic on expectation in the reactivation-extinction model. For example, in human studies, Yang et al. (2019) asked exactly where prediction error is critical in reconsolidation (PMID: 31585344). Li et al. (2019) further used fMRI to look into the related brain mechanisms (PMID: 31669978). Relevant animal studies can be found in PMID: 30659275 and PMID: 29809041.

There are also unfortunately concerns around the effectiveness of the manipulation. First, the 3 groups were not homogenous at acquiring the differential conditioning based on the FSR measurement. Second, the critical manipulation on expectation for learning did not lead to significant difference between groups 1 and 2. Third, a lack of CS+ and CS- differentiation at early extinction on day 2 in both measurements casts doubts on why no memory of the differential conditioning. Fourth, it is unclear whether the shock intensity that participants decided was comparable across groups to avoid confounding. Given these concerns, it is unclear whether the findings can be unambiguously interpreted for drawing conclusions on the research question.

6. PLOS authors have the option to publish the peer review history of their article (what does this mean?). If published, this will include your full peer review and any attached files.

Reviewer #1: No

Reviewer #2: No

---

## [Author Response · Author response to Decision Letter 0]

19 Aug 2021

Thank you providing feedback on our manuscript. We have carefully reviewed your comments and have made changes based on your feedback. We hope that these changes address the concerns that you have raised. Below you will find a detailed response to each query (in italics).

1. Is the manuscript technically sound, and do the data support the conclusions?

Reviewer #1: No

Reviewer #2: No

Please see individual comments to each reviewer below.

2. Has the statistical analysis been performed appropriately and rigorously?

Reviewer #1: Yes

Reviewer #2: Yes

Thank you. 

3. Have the authors made all data underlying the findings in their manuscript fully available?

Reviewer #1: No

Reviewer #2: No

We contacted our research ethics board to inquire as to whether we are able to share the anonymized dataset publicly. Unfortunately, we are not able to make the data set public because participants only consented to the research team having access to the data. We have adjusted our consent procedures recently and now inform participants that their anonymized data may be made available as a publicly accessible dataset to support open science. Unfortunately, during the collection of the data for this study, we had not yet implemented these procedures.

4. Is the manuscript presented in an intelligible fashion and written in standard English?

Reviewer #1: No

Reviewer #2: Yes

Please see our comments for each reviewer below.

5. Review Comments to the Author

Reviewer #1: The current research investigated whether expecting to learn something during exposure enhances reconsolidation of fear memory. While this research question may be relevant to understand the boundary conditions of reconsolidation, several conceptual and methodological issues arise that limit interpretation of the current findings. Most importantly, the failed manipulation and fear acquisition, and unclear statements in the introduction and discussion section greatly devalue the manuscript.

Title:

- Unclear what is meant by ‘the expectation’

We have changed the title of our manuscript to: Can learning expectancy enhance reconsolidation using the post-retrieval extinction paradigm? to reflect terms that have been used in previous research on learning expectancy and reconsolidation (e.g., Sevenster, Beckers, & Kindt, 2012). Given space limitations for titles (and abstracts) it is challenging to provide a clear definition of learning expectancy. We hope that the introduction and methods help clarify the definition. 

Sevenster D, Beckers T, Kindt M. Retrieval per se is not sufficient to trigger reconsolidation of human fear memory. Neurobiol Learn Mem. 2012;97(3):338–45. https://doi.org/10.1016/j.nlm.2012.01.009.

Abstract:

- The objective is unclear. What do the authors mean with ‘expectation for learning’?

- The study took place ‘on’ three consecutive days.

- The explanation of the manipulation is unclear for two reasons. First, what are the oral instructions? Second, it is hard to follow since memory reactivation was not mentioned yet.

- The results start with the interpretation of the findings. I would recommend merely reporting the findings at this point.

- No keywords are reported.

Thank you for the feedback. We have incorporated all the above feedback in the abstract on page 2, most notably revising our description of the methods and the results and have added key words. 

Introduction:

- Two sentences barely form a paragraph (first paragraph). Please integrate this with the following paragraph.

We have merged the first and second paragraphs of the introduction together (see page 3). 

- Eliminate the word ‘consistently’, because in this sentence it is clear that these are not consistent findings: “Studies in humans have consistently demonstrated that a conditioned fear response, as measured by fear potentiated startle (FPS), can be eliminated if reactivation is paired with oral administration of propranolol [4, 7, 8, 9], though a few studies have failed to replicate this effect[10, 11].”

We have removed the word “consistently” from this sentence (see page 3). 

- In the following sentence the authors refer to ‘these studies’, yet they only discuss one study in that paragraph. This is somewhat confusing: “These studies illustrate that reactivated memories following the administration of propranolol, can be modified and this process might be beneficial in the treatment of anxiety based psychological disorders.”

By incorporating the first two paragraphs, we have now deleted this sentence (see page 3). 

- Why does propranolol only indirectly target the mechanisms involved in reconsolidation? Please explain.

We have clarified why propranolol indirectly targets reconsolidation (see page 4) .

- The authors only briefly touch upon the issues in reconsolidation research. They merely state that inconsistencies in replicating these findings suggest that memory recall may not be sufficient. However, there is already a large body of potential boundary issues that are not discussed. Also, in the work by Schiller, many participants were removed prior to data analyses. This could also explain failed replications later on, but this is not discussed by the authors.

When this study was originally developed the replication findings were not yet published. We have now included a discussion of the failed replication of Schiller’s work at the bottom of page 4. For space limitations we have not discussed other boundary conditions beyond the expectation for learning and prediction error, though we have expanded our discussion of those processes (see page 6). 

Methods:

- For the power analysis, it is unclear what tests the power analysis was based on. Moreover, is a medium effect the effect that the authors expect? On which studies was that effect size based as there are so many failed replications already?

On page 14, we now provide more detail on the estimated used for our power analysis. 

- What were the criteria for usable physiological data? Later on, the authors state that 12 participants were excluded because their physiological data was not readable. It remains unclear what the exclusion criteria were.

Unreadable data was defined as excessive noise based on visual inspection or evidence of clipping (e.g., data that was outside of the recording range). We have clarified this on page 8.

- Why did 33 participants drop out? That was one third of your sample.

This is a good question and something that we wondered as well. We believe the main reasons people dropped out of the study was due to our recruitment method and study schedule. We recruited undergraduate participants from the Integrated System (ISPR) at the University of Ottawa. Participants could only receive two credits total (i.e., one credit for day one and one credit for day two) for their participation in the study. However, participants were required to participant on all three days. Most participants were unaware of the third day of the study requirement (though it was indicated in both the recruitment material and consent form), only required two credits, or had scheduling conflicts which prevented them from participating on all three days. For conciseness we have not included this information in the manuscript, however, we would be happy to do so if the reviewer feels it would be important to add.

- What was the reinforcement rate? ‘Sometimes paired with a shock’ is vague.

We now provide information on the reinforcement rate on 8 of the manuscript. 

- Inconsistency in order when describing STAI-T and STAI-S.

We have revied our description of the STAI-T and STAI-S and have made sure that we present findings for the STAI-T first and the STAI-S second. We hope that this is what the reviewer was requesting and if not, further clarification would be helpful to further address this concern.

- Why was the manipulation check retrospectively measured? The answers may be influenced by what actually happened.

 We have expanded on our rationale for conducting the manipulation check retrospectively on page 9-10.

- The authors pose that SCR is a measure of anxiety. Yet, some authors claim that fear and anxiety are different constructs. In addition, some researchers state that SCR measures arousal and not fear per se. Can the authors please elaborate on that?

We concur that there is considerable debate as to the exact distinction between fear and anxiety. Additionally, there is discussion concerning whether SCR is related to an orienting response or arousal, given that SCR increases in response to positive as well as negative stimuli (e.g., Cacioppo, Tassinary, & Berntson, 2007; Hamm & Weike; 2005). Given the fact that SCR and FPS are frequently used as an outcome measures in fear conditioning studies (see reviews by Beckers et al., 2013; Ryan et al., 2019) we have removed the descriptions of what FPS and SCR are believed to measure from the method section (see page 10). However, we have expanded our discussion section further, and discuss what each of these physiology measures are believed to reflect on page 23-24.

Beckers T, Krypotos AM, Boddez Y, Effting M, Kindt M. What's wrong with fear conditioning? Biological Psychology. 2013 Jan;92(1):90-96. DOI: 10.1016/j.biopsycho.2011.12.015.

Cacioppo, J. T., Tassinary, L. G., & Berntson, G. G. (Eds.). (2007). Handbook of psychophysiology (3rd ed.). Cambridge University Press. https://doi.org/10.1017/CBO9780511546396

Hamm AO, Weike AI. The neuropsychology of fear learning and fear regulation. Int J Psychophysiol. 2005 Jul;57(1):5-14. doi: 10.1016/j.ijpsycho.2005.01.006. Epub 2005 Apr 21. PMID: 15935258.

Ryan KM, Zimmer-Gembeck MJ, Neumann DL, Waters AM. The need for standards in the design of differential fear conditioning and extinction experiments in youth: A systematic review and recommendations for research on anxiety. Behav Res Ther. 2019 Jan;112:42-62. doi: 10.1016/j.brat.2018.11.009. Epub 2018 Nov 20. PMID: 30502721.

- EMG abbreviation is not written in full.

We have now indicated that EMG stands for electromyography on page 10. 

- The authors make a strong statement about what FPS represents:

“This measure is used to assess the startle response of the participant as neurologically it represents the connections from the amygdala to the startle-reflex pathway in the brainstem [35].”What do the authors mean by this?

As mentioned above, we have opted to not to discuss the hypothesized meaning of FPS and SCR, and have therefore deleted this statement from page 10. A more detailed description on what FPS and SCR are believed to reflect can be found in the discussion section on page 23-24. 

- Participants were connected to the skin conductance à incorrect language, please rephrase.

We have now clarified the wording of the procedures on page 12. 

- What was the duration of the CS presentation, US presentation, and intertrial intervals?

We have now included information on the duration of the CS and US, as well as information on intertrial intervals on page 11.

- Terminology is not used, such as contingencies and reinforcement rate.

We have refined the terminology used throughout the method section.

- Are there no noise alone trials measured for FPS?

At the start of each session there were 10 habituation trials of noise alone. This information is outlined at the end of the first paragraph of the procedure section (see page 11). We also provide reminders of this in our description of what happens on each of the days.

- “This break allows for the activation of the neural mechanisms needed for reconsolidation to take place [1].” à Unclear what the authors mean by this.

We deleted this sentence on page 13. 

- It is only indirectly stated what the memory reactivation was, namely: “Participants in the No

Reactivation condition were not exposed to the single presentation of the CS+ (i.e., their

condition fear memory was not reactivated) and instead proceed straight to the 10-minute break.”

Please make this more explicit.

We thank the reviewer for their feedback on improving the clarity of our description of procedures. We now include a statement explicitly defining reactivation on page 12.

- How many extinction trials were there?

To ensure equivalent exposure to the CS+ and the CS- on day 2, participants in the no reactivation group were shown 11 presentations of both the CS+ and CS-, whereas participants in the reactivation conditions (e.g. the Expectation for learning and the No expectation for learning groups), who were previously presented with one trial of the CS+ during reactivation, were given 10 presentations of the CS+ and 11 presentations of the CS-. This information can now be found on page 13.

- What was the order of trials in the re-extinction (or reinstatement) phase? Which CS was presented first? And was the order counterbalanced across participants?

The CS+ was always presented first. Participants then received 9 CS+ trials and 10 CS- trials in pseudorandomized order with half of the participants receiving a CS+ presentation on the second trial and the other half receiving a CS- presentation on the second trial.. We have clarified the details of re-extinction phase on page 13. 

- Why did participants in the no reactivation group did not receive the manipulation check? Obviously it matters whether the other two groups differs from each other, but also whether they differ from the control group. That would give more direct information about the necessity for prediction error.

We agree that this would have provided helpful information and should be included in future studies. However, because we did not manipulate prediction error in the no reactivation group, we did not use a manipulation check for this group . We have outlined this as a limitation in our discussion of the manipulation check on page 22.

- Statistical assumptions were violated, but only the correction for normality violation is reported here. What were the other violations and how is that corrected?

In addition to the assumption of normality, we also examined homogeneity of variance using FMax, and have noted this on page 14. Homogeneity of variance was violated at only one time point. Though this limits interpretation of the overall ANOVA, the follow-up tests focus on within-participant comparisons (paired t-tests) and therefore should not be affected. We note the violation page 18. We did not test for sphericity was there were only two levels of each within participant factor.

- A lot of data is disregarded by the statistical approach (e.g., first and late acquisition phase). That is quite wasteful. Why was this approach chosen?

Given the number of trials an ANOVA looking across all trials is quite complex and poses difficulties for interpretation. Additionally, given the degree of variability in a single response trial we felt that taking average responses for the beginning and end of acquisition and extinction would offer a more reliable indicator of whether acquisition and extinction took place.

Results

- A conclusion from the manipulation check is lacking, but if I understood correctly, the manipulation failed. The No Expectation for Learning condition reported that they still expected to receive a shock and this expectation did not differ from the Expectation for Learning group. Thus, the study cannot answer the research question. 

We agree with the reviewer that failed manipulation of expectancy limits our ability to draw conclusions about the initial planned research question. We have expanded our discussion of on page 23. However, we feel that this failed manipulation is important information in and of itself and expand our discussion about why we suspect the manipulation failed. We believe this is important information to share with researchers and can contribute to extending our understanding of learning expectancy and reconsolidation.

- Did the authors not expect a difference at the first extinction trial? As the intervention was already before the extinction phase it makes sense that the No Expectation for Learning condition displayed reduced conditioned responding compared to the other groups on the first trial in this phase. Yet, since the manipulation was not successful, obviously there were no group differences. 

Since the manipulation was expected influence reconsolidation rather than extinction learning, we did not expect that there would be differences at the start of extinction. We report on extinction trials to establish that there was a reduction in physiological responding to the CS+ during the updating phase. We have further specified on page 17 that we predicted that extinction would be apparent across all groups.

- In addition, results cannot be interpreted since fear acquisition on FPS was unsuccessful (except in the No Expectation for Learning group. Why did the authors analyze the extinction and reconsolidation phase if there was no learning in the first place?

We thank you for this important comment. We decided to proceed with our a priori analytic strategy, and to broadly include all participants that at least demonstrated fear acquisition based upon self-report (e.g., they successfully identify the contingencies), rather than physiological responding. This maximizes the number of participants included and also makes our sample more broadly generalizable. We recognize that the choice of inclusion and exclusion criteria, as well as the definition of what successful learning looks like has a big impact on results. We have added a discussion concerning this issue on page 24-25. 

Discussion

- The authors conclude that only in the Expectation for Learning group CS+ responding remained stable following reinstatement. Yet, this is not the case when only the participants with successful fear acquisition were included, since also in the No Expectation for Learning group FPS remained stable. Thus, this conclusion is incorrect. Although the authors state that these findings or not robust and that this effect disappeared when only including participants with successful fear acquisition, the conclusion should be that in two groups CS+ responding remained stable.

Thank you for raising this important point. To improve the clarity of findings we now report paired t-tests for the post-hoc analysis for each group. This provided the reader with a sense of both the number of participants who showed fear acquisition across each groups (the df + 1), and demonstrates that the pattern of response in each group was similar in the smaller sample as it was for the larger sample. 

Furthermore, we have significantly toned down our discussion concerning our interpretation of our findings throughout the discussion. Nonetheless we do feel that two important issues are raised from our null findings: a) verbal manipulations of expectancy may not be adequate to affect reconsolidation and b) that careful attention needs to be made when measuring SCR and FPS simultaneously. We have opted to focus the discussion on these two factors. 

- The authors start explaining the dissociation in their data. First of all, there seems to be no dissociation as they did not find any results on both SCR and FPS. Second, with this paradigm, the interventions target reconsolidation of threat memories, not of SCR itself. As a result of successful reconsolidation, SCR responding should be lower. Therefore, the following sentence seems incorrect: “Researchers have suggested that SCR reflects a cognitive representation of arousal and is more difficult to reconsolidate”. Finally, as stated before, the dissociation (or absence of dissociation) between different physiological responses is highly debated. Stating that SCR reflects a cognitive representation of arousal is unclear and seems incorrect.

We thank the reviewer for their feedback. We agree that the focus of this research is on the reconsolidation of fear conditioned memories, not SCR or FPS. Nonetheless the majority on the reconsolidation of fear conditioned memories do use SCR and/or FPS as proxies for fear.

We respectfully disagree that there was no dissociation in our results. For example, there was evidence of fear conditioning during acquisition based using SCR but not FPS. Our sentence that you refer to may be unclear and we have therefore substantially revised that section of discussion and provide additional support for some of the statements we make concerning the the distinction between SCR and FPS (see page 23 – 24). We hope that we have better clarified the point we were trying to make

- The paragraph on loud tones interfering with SCR does not make sense. Obviously loud sounds can interfere with SCR, but there is a plethora of studies demonstrating that FPS and SCR can simultaneously be measured. Moreover, it remains unclear how this would explain an absence of effects, since the expected findings were also absent on FPS.

Though we are not the first researcher to make this suggestion (e.g., Lissek et al., 2005). Nonetheless, we do agree with the reviewer that many studies have used FPS and SCR simultaneously. We have removed this section of the manuscript in light of some of the other changes that we have made to our discussion on FPS and SCR (e.g., pages 23-24).

Lissek S, Baas JM, Pine DS, Orme K, Dvir S, Nugent M, Rosenberger E, Rawson E, Grillon C. Airpuff startle probes: an efficacious and less aversive alternative to white-noise. Biol Psychol. 2005 Mar;68(3):283-97. doi: 10.1016/j.biopsycho.2004.07.007. PMID: 15620795.

- The limitations of physiological responding are not properly discussed. How would the authors explain that there was differential responding in fear acquisition, but that it just was not visible until day 2?

We thank the reviewer for this point. We initially based this statement upon visual inspection of the graph. However, we have now computed paired t-tests to examine this statistically. Though, FPS was indeed grater at the start of extinction compared to the end of acquisition for both the CS+, t (47) = 3.2, p = .002, and CS-, t (47) = 2.59, p = .01, the difference between the start of acquisition and start of extinction was not meaningful for either the CS+, t (47) = 1.04, or the the CS-, t (47) = .28, p = .78. This suggests to us, that participants did not acquire fear as measured by FPS. We have therefore removed that sentence from our discussion (page 24), and instead focus on the following a) the patterns were small but in the expected direction; b) that we defined fear acquisition based on contingency awareness, and c) the importance of considering exclusion criteria throughout the discussion.

- Why is the failed manipulation check only discussed so late in the discussion section? This is a major problem with this study. The authors claim that the manipulation check may not have measured whether the participants expected to learn something. Even if that is the case, it can be argued that participants in both the Expectation for Learning and No Expectation for Learning were learning something (namely that they did not receive shocks, although they expected these). The same holds for the No Reactivation group. Is there any reason to believe that they had different expectations about whether they would learn something?

Thank you for this point and we agree that the note about the failed manipulation should be discussed earlier in the discussion. We moved the discussion about the failed manipulation check to the second paragraph in the discussion (beginning on page 21), and focus more upon understnding our failure to manipulate the expectation for learning in light of previous research. It is possible that the no reactivation group was learning something as well (though presumably not under conditions that would induce reconsolidation). We regret that we did not ask them the manipulation check question as well, and have noted this as a limitation on page 22.

- There is quite some literature suggesting that verbal responses are quite strong to change expectancies (e.g., Mertens et al., 2020), and actually change fear responding.

Thank you for this interesting article. Upon reading it, we incorporated important details on page 23 regarding verbal contingency instructions on fear conditioning. 

APA:

- Schiller et al., [1]. The comma should be omitted.

- Spelling mistakes, such as “i.e., Post-reterival extinction paradigam”, “propanrolol”

- Multiple double spaces

- Language: the manuscript is not densely written and feels quite lengthy.

- Biased language, for instance “62% female”

- Inconsistent use of capital letters, for instance for group names.

We have noted these errors and have updated the manuscript accordingly.

Refererences

Mertens, G., Boddez, Y., Krypotos, A. M., & Engelhard, I. M. (2020). Human fear conditioning is moderated by stimulus contingency instructions. Biological Psychology. https://doi.org/https://doi.org/10.1016/j.biopsycho.2020.107994

Reviewer #2: The authors used a retrieval-extinction paradigm to ask the role of expectation in reconsolidation. They recruited 3 groups of 48 undergraduate students who were presented with a spider image that was paired with a shock and another image was not paired (i.e. differential fear conditioning). They were randomly assigned to group 1: reactivation with expectation for learning, group 2: reactivation without expectation, or group 3: no reactivation before they all received an extinction session. On the next day, they received unpaired shocks and further extinctions trials. They found no difference in expectation in the 2 reactivation groups. Based on the SCR measurement, groups 1 and 2 both showed fear return for both CS+ and CS- after reinstatement while group 3 showed fear return for CS+. Based on the FPS measurement, groups 1 and 2 both showed fear return for both CS+ and CS- after reinstatement.

Although this study addresses a subject of importance, it did not cite reference already addressing this specific topic on expectation in the reactivation-extinction model. For example, in human studies, Yang et al. (2019) asked exactly where prediction error is critical in reconsolidation (PMID: 31585344). Li et al. (2019) further used fMRI to look into the related brain mechanisms (PMID: 31669978). Relevant animal studies can be found in PMID: 30659275 and PMID: 29809041.

There are also unfortunately concerns around the effectiveness of the manipulation. First, the 3 groups were not homogenous at acquiring the differential conditioning based on the FSR measurement. Second, the critical manipulation on expectation for learning did not lead to significant difference between groups 1 and 2. Third, a lack of CS+ and CS- differentiation at early extinction on day 2 in both measurements casts doubts on why no memory of the differential conditioning. Fourth, it is unclear whether the shock intensity that participants decided was comparable across groups to avoid confounding. Given these concerns, it is unclear whether the findings can be unambiguously interpreted for drawing conclusions on the research question.

Thank you for your helpful feedback. Many of the concerns you have raised are mentioned in our feedback to reviewer 1.

Thank you for the additional articles. We have now included reference to these articles in the introduction (page 6-7) and discussion (page 22).

We agree with reviewer 2 that the failed manipulation limits our ability to draw conclusions about the impact of learning expectancy on reconsolidation. Nonetheless, as outlined above we believe the failed manipulation provided informative information for future research (see page 22-23). 

We also agree that our inability to demonstrate differential fear conditioning using FPS is also problematic. Again, we expand our discussion on why we think this may be the case (see page 22-23).

As noted above, we have significantly scaled back our discussion regarding our findings for the main research question, and instead focus on some of the informative findings regarding expectancy manipulation and the measurement of FPS and SCR in the context of fear conditioning. 

We are unclear about the reviewer’s reference to a lack of CS+ and CS- differentiation at early extinction on day 2, as we demonstrate that there was greater SCR and FPS to the CS+ and CS- on page 17-18, and 19-20 respectively.

We have now demonstrated that the voltage selected across our groups was similar (see page 16).

6. PLOS authors have the option to publish the peer review history of their article (what does this mean?). If published, this will include your full peer review and any attached files.

Do you want your identity to be public for this peer review? For information about this choice, including consent withdrawal, please see our Privacy Policy.

Reviewer #1: No

Reviewer #2: No

---

## [Decision Letter · Decision Letter 1]

15 Feb 2022

PONE-D-20-33271R1Can learning expectancy enhance reconsolidation using the post-retrieval extinction paradigm?PLOS ONE

Dear Dr. Ashbaugh,

Thank you for submitting your manuscript to PLOS ONE. After careful consideration, we feel that it has merit but does not fully meet PLOS ONE’s publication criteria as it currently stands. Therefore, we invite you to submit a revised version of the manuscript that addresses the points raised during the review process.

We look forward to receiving your revised manuscript.

Kind regards,

Marta Andreatta

Academic Editor

PLOS ONE

Journal Requirements:

Additional Editor Comments:

Dear Ms. Ashbaugh,

three experts of the field reviewed your article and all of them pointed out that it could be interesting but many aspects are not sufficiently addressed for replication.

I invite you to answer their valueable comments carefully and, as pointed out especially by Reviewer 1, to update the literature.

I also read your manuscript and I noticed several missing information such as a precide description of the pre-processing for the physiological indices. The introduction focuses on pharmacological manipulation and it does not present much information about the effects of instrucitons, which is more the focus of your study. Despite the power analysis, I think the three groups of 16 participants each are small, especially in consideration of these times of replication crises, and this should be made clear in the abstract as well as in the manuscript.

Best regards,

Reviewers' comments:

Reviewer's Responses to Questions

**Comments to the Author**

1. If the authors have adequately addressed your comments raised in a previous round of review and you feel that this manuscript is now acceptable for publication, you may indicate that here to bypass the “Comments to the Author” section, enter your conflict of interest statement in the “Confidential to Editor” section, and submit your "Accept" recommendation.

Reviewer #3: (No Response)

Reviewer #4: (No Response)

Reviewer #5: (No Response)

2. Is the manuscript technically sound, and do the data support the conclusions?

Reviewer #3: Yes

Reviewer #4: No

Reviewer #5: Partly

3. Has the statistical analysis been performed appropriately and rigorously? 

Reviewer #3: Yes

Reviewer #4: Yes

Reviewer #5: Yes

4. Have the authors made all data underlying the findings in their manuscript fully available?

Reviewer #3: No

Reviewer #4: No

Reviewer #5: No

5. Is the manuscript presented in an intelligible fashion and written in standard English?

Reviewer #3: Yes

Reviewer #4: Yes

Reviewer #5: No

6. Review Comments to the Author

Reviewer #3: In the present study entitled ‘Impact of the expectation on memory reconsolidation using a post retrieval extinction paradigm’, by Marinos and Ashbaugh, authors aimed to investigate the effects of the expectation for learning on reconsolidation of conditioned fear memories using the post-retrieval extinction paradigm in humans. For this purpose, 48 healthy participants underwent a fear conditioning experiment that took place over three consecutive days. The expectation for learning was manipulated giving instructions prior to memory reactivation; fear potentiated startle (FPS) and skin conductance response (SCR) were taken as implicit measures of fear. Results showed a small effect of the expectation for learning on reconsolidation with FPS as a measure of fear, but no evidence of reconsolidation was observed for SCR. Authors conclude stating that a verbal manipulation of the expectation for learning may not be salient enough to induce reconsolidation as measured by SCR but may be sufficient as measured by FPS.

In general, I think the idea of this article is interesting and the authors’ fascinating observations may be of interest to the readers of Plos One. However, some comments, as well as some crucial citations that should be included to support the authors’ argumentation, need to be addressed to improve the article, its adequacy, and its readability prior to the publication in the present form.

Comments

• Regarding the Abstract: I think that the lack of an explanation of what “expectation for learning” means in this study makes the reader unable to grasp the key aspects of this paper by consulting the abstract. I suggest reorganizing the abstract, making sure to include an explanation of this concept. Also, keywords are missing.

• Regarding the Introduction: In general, I strongly recommend authors to reshape this section, which I believe is way too much simplistic to outline a comprehensive yet rigorous investigation of the subject of this study (i.e., memory reconsolidation). Thus, I suggest the authors to make such effort to provide a brief overview of the pertinent published literature that offer a more in depth as well scientific perspective on reconsolidation, to provide a more defined background.

• Page 2: correct a typo, change “enhanced” in “enhance”.

• Page 3-6, Introduction: Authors provided a detailed overview on the different methods used to interfere with the reconsolidation process, only focusing on pharmacological interventions. However, the literature they cite is neither exhaustive nor up to date. Still, I think that adding some studies that could provide further insight on different techniques (i.e., Non-invasive brain stimulation techniques, NIBS) used to study reconsolidation would be crucial in this section: for example, Borgomaneri and colleagues’ study (2020, Current Biology - https://doi.org/10.1016/j.cub.2020.06.091) showed that the inhibition with of the dorsolateral prefrontal cortex (DLPFC) with Transcranial magnetic stimulation (TMS) after emotional memory reactivation disrupts physiological responding to learned fear, highlighting the role of this area in the neural network that mediates the reconsolidation of fear memories in humans. I also suggest mentioning a review by Borgomaneri and colleagues (2021, Neuroscience and Biobehavioral Reviews - https://doi.org/10.1016/j.neubiorev.2021.04.036) that specifically focused on human reconsolidation and on the ability of non-invasive brain simulation (NIBS) to interfere with activity of neural circuits (i.e., amygdala-mPFC-hippocampus) involved in the acquisition and reconsolidation of emotional memories. Finally, I would suggest one of the latest Borgomaneri and colleagues’ study (2021, Journal of Affective Disorders - https://doi.org/10.1016/j.jad.2021.02.076), that illustrated the therapeutic potential of NIBS as a valid alternative in the treatment of abnormally persistent memories that characterized those patients with anxiety disorders that do not respond to psychotherapy and/or drug treatments, promoting reduction of fear by focusing on the reconsolidation process. Moreover, if the authors consider it appropriate, they can also see additional studies, for example Censor and colleagues’ study (2014, Cortex - https://doi.org/10.1016/j.cortex.2013.05.013) or Sandrini and colleagues’ study (2018, Frontiers in psychology – https://doi.org/10.3389/fpsyg.2018.01430) that provide a general overview on the topic.

• Page 4: please add “to” after “translational impact”.

• Page 4: correct a typo, change “sufficient” to “sufficient”.

• Page 5: correct a typo, change “propanrolol” to “propranolol”.

• Page 6: The hypothesis of the study is clearly presented and adequately outlined; however, authors should consider adding a detailed definition of what the expectation of learning, as it is the variable that is manipulated in the study.

• Page 9: correct a typo, change “participates” in “participants”.

• Page 8, Unconditioned stimulus (US): please provide the mean shock intensity for each group.

• Page 10: specify the acronyms “FPS” after the full name “Fear potentiated startle”.

• Regarding the Methods: What were the criteria adopted for usable physiological data? Authors stated that 12 participants were excluded because their physiological data was not readable. Please explain in detail what is meaning ‘not readable’ and use the proper study to exclude data from the dataset collected.

• Regarding the Results: I suggest rewriting this section more accurately. I think it is misleading to say that results show a dissociation between SCR and FPS. This likely happened because the ‘Expectation for Learning’ group failed to show fear acquisition as measured with FPS, thus showing no difference after reinstatement is to be expected. Moreover, authors decided to run an analysis including only participants that showed successful fear acquisition, resulting in no difference among groups. This idea must be rewritten since it is misleading and does not properly convey the results found.

• Page 19, Post hoc analysis: authors should specify what post hoc test they decided to conduct.

• Page 20, Discussion: Authors stated that they found ‘no evidence of reconsolidation’. Reconsolidation takes place every time an individual is shown a reminder of a previously formed memory. What authors could say is instead that the experimental manipulation was not successful in altering this process, thus resulting in the memory being evoked as originally consolidated on day 1. I suggest rephrasing this concept.

• Regarding the Discussion: In my opinion, this paragraph would benefit from some thoughtful as well as in-depth considerations by the authors, because as it stands, it is very descriptive but not enough theoretical as a discussion should be. Authors should make an effort, trying to explain the theoretical implication as well as the translational application of their research. Also, I suggest reorganizing the final part of this section, because I do not think that the results suggest that verbal manipulations of the expectations for learning do not induce reconsolidation, but rather that they are not enough to manipulate said process. Please reshape this section so they can better represent this concept and include the final statements in a ‘Conclusion’ paragraph, to summarize key points and elucidate possible keys to advancing research and understanding of the reconsolidation process.

• Although the Authors have acknowledged the study’s limitations at the end of the manuscript, I also suggest including a properly defined ‘Limitations and future directions’ subsection, in which they can describe in detail and report all the technical issues brought to the surface and discuss theoretical and methodological avenues in need of refinement.

• Regarding the Figures: I suggest adding images of the stimuli and of the experimental design used in the experiment. In my opinion, this visual representation of the experimental procedure will dramatically improve the study's readability and comprehension.

Reviewer #4: Abstract

Objective: the background to the study is not adequately described here

Introduction: In the introduction a lot is written about studies with propranolol, whereby the pharmacological manipulation of the reconsolidation is not the subject of this paper.

The introduction about studies trying to replicate Schillers original results is missing the Paper of Astana et al., (2016), indicating individual differences in Val66Met polymorphism might modulate the effect of reconsolidation.

Page 5: typing error: ezamined

Materials:

The details about the SCR and Starle analyses are totally missing. Please describe in detail the filter, artefact detection approaches, the time windows to define a peak, baseline correction, normalization of the data.

Is it correct, that no participants had to excluded due to missing fear acquisition?

Results

Figure 2 and 4 are not optimal to see the relevant effects.

Literature:

Asthana MK, Brunhuber B, Mühlberger A, Reif A, Schneider S, Herrmann MJ. Preventing the return of fear using reconsolidation update mechanisms depends on the met-allele of the brain derived neurotrophic factor Val66Met polymorphism. International Journal of Neuropsychopharmacology. 2016; 19(6):1–9. DOI: 10.1093/ijnp/pyv137

Reviewer #5: The general approach to use a verbal instruction is interesting regarding potential therapeutic options and to (theoretically) contrast it with pharmacological interventions.

In their point-to-point response the authors addressed the concerns of the reviewers and revised the manuscript substantially. However, reading the manuscript and the responses to the reviewers, I am not convinced that the main concerns we fully addressed. In the current form, the manuscript still lacks some important information or - if given - does not always convey it in an understandable way. There are also several minor language/style/spelling(comma issues that the authors should take care of (to name a few, e.g. “these drugs, are toxic to humans …”, “been replicated [17, 18] however; several …”, “boudnary … ”, “CSa+” in the figure legend).

I will state my major concerns in the following. Before that, I would like to say that results are how they are. If something does not speak in favor of a hypothesis, it should be extensively discussed as this improves scientific research.

Major points:

1. Lack of data availability: I am not sure how the Journal handles such issues, although they recommend data to be available to other researchers

2. Presentation of procedures/results: The manuscript would benefit a lot from a clearer description of procedures and results. A schematic figure / illustration would be an option. Then, the reader could better grasp the results when having a look at the procedure then.

3. Related to my second points are some parts of the analysis description that I find hard to understand. For example: “For extinction, the early phase of extinction was calculated by taking the averages of trials one and two on day two. The late phase of extinction was calculated by taking the averages of trials 10 and 11 on day two.” This is one way, but in your specific design, I have some problems to apply this to the three groups. Two groups did 1 CS+ and then after the 10 minutes 10 CS+ and 11 CS-, whereas the third groups did 11 CS+ and 11 CS-. Does this mean that one CS+ of the early extinction process was administered 10 earlier than in the other group (see figures 1 and 3, and page 13, where Extinction trial 1 has only a CS-). In figure 1 or 3: Where is the additional CS+ for the third group? Is it included in the reactivation trial of the other two groups.

4. Unwarranted result presentation/conclusions. For example, see the abstract section with the results (“…, and increased in response to the CS+ but not the CS- in the no reactivation group.”). This is actually not the case as this is a result that comes from a posthoc analysis after a non-significant interaction with group. This is not proper statistics. You should only use posthoc analysis if you have significant main and/or interaction effects. Having a closer look at the results shows the following (“The no reactivation group showed an increase in their SCR to the CS+, t (15) = -2.47, p = .03, d = -.84, but not to the CS-, t (15) = -1.86, p = .08, d = -.58, from the end of extinction on day two to the beginning of re-extinction on day three.”). Two things are apparent. First, the phrase “but not to the CS-” results from a p value of .08, with an ES of about .6. The significant comparison “to the CS+” results from a p value of .03, with an ES of about .8. This looks as if one significant posthoc effect and one nonsignificant posthoc effect are interpreted as being different from each other (no interaction).

5. Differences in procedures for the groups. It is suboptimal that there are differences between the groups. For example, on day 2, the third group has a different timing as they did not get a reactivation trial. It might have been better to have a reactivation trial for them, too, but without any verbal instruction. The same is true for the manipulation task, as was already pointed out by the other reviewers. Because of the reactivation for the tow groups they have a 10 CS+/11 CS- extinction phase, the third group 11 CS+/11 CS- extinction phase.

6. Low Number of participants per group and dropouts: This should be properly discussed.

7. The title should not contain a question. It could also state a null finding.

7. PLOS authors have the option to publish the peer review history of their article (what does this mean?). If published, this will include your full peer review and any attached files.

Reviewer #3: No

Reviewer #4: No

Reviewer #5: No

---

## [Decision Letter · Decision Letter 2]

12 Jul 2022

Verbal manipulations of learning expectancy do not enhance reconsolidation

PONE-D-20-33271R2

Dear Dr. Ashbaugh,

We’re pleased to inform you that your manuscript has been judged scientifically suitable for publication and will be formally accepted for publication once it meets all outstanding technical requirements.

Kind regards,

Dario

Dario Ummarino, PhD

Senior Editor

PLOS ONE

Additional Editor Comments:

As you can see in the final reviewer report attached below, the reviewer requested that an additional reference is added to your manuscript. Please consider this request as optional. Also, we noted that a previous request to cite reference 24 in your current manuscript may have not been necessary, and therefore we suggest that you consider removing this citation, if you prefer.      

Reviewers' comments:

Reviewer's Responses to Questions

**Comments to the Author**

1. If the authors have adequately addressed your comments raised in a previous round of review and you feel that this manuscript is now acceptable for publication, you may indicate that here to bypass the “Comments to the Author” section, enter your conflict of interest statement in the “Confidential to Editor” section, and submit your "Accept" recommendation.

Reviewer #3: All comments have been addressed

2. Is the manuscript technically sound, and do the data support the conclusions?

Reviewer #3: Yes

3. Has the statistical analysis been performed appropriately and rigorously? 

Reviewer #3: Yes

4. Have the authors made all data underlying the findings in their manuscript fully available?

Reviewer #3: Yes

5. Is the manuscript presented in an intelligible fashion and written in standard English?

Reviewer #3: Yes

6. Review Comments to the Author

Reviewer #3: I am very pleased to see that the Authors have welcomed my suggestions and have clarified most of the questions I raised in my first round of this review. I believe that this original research article does an excellent work describing how a verbal manipulation of the expectation of learning may not be salient enough to enhance reconsolidation of fear memories.

I only have few last minor suggestions, to further improve the theoretical background of the present article and its argumentation on neural basis and physiological mechanisms of fear conditioning / learning.

For this reason, I would suggest adding a new theorethical manuscript that provides an overview of the anatomical–functional interplay between the prefrontal cortex and heart-related dynamics in human fear conditioning and proposes a theoretical model to conceptualize psychophysiological processes, published on Trends in Neurosciences. (https://doi.org/10.1016/j.tins.2022.04.003).

Overall, this is a timely and needed study, and I look forward to seeing further study on this issue by these authors in the future.

I look forward to seeing further study on this issue by these authors in the future.

Thank You for your work.

7. PLOS authors have the option to publish the peer review history of their article (what does this mean?). If published, this will include your full peer review and any attached files.

Reviewer #3: No

---

## [Editor Report · Acceptance letter]

9 Aug 2022

PONE-D-20-33271R2 

Verbal manipulations of learning expectancy do not enhance reconsolidation 

Dear Dr. Ashbaugh:

I'm pleased to inform you that your manuscript has been deemed suitable for publication in PLOS ONE. Congratulations! Your manuscript is now with our production department. 

Kind regards, 

on behalf of

Dr Dario Ummarino, PhD 

Staff Editor

PLOS ONE